# HARD: Hyperplane ARrangement Descent

Tianjiao Ding[1*]    Liangzu Peng[2*]    René Vidal[2]

The problem of clustering points on a union of subspaces finds numerous applications in machine learning and computer vision, and it has been extensively studied in the past two decades. When the subspaces are low-dimensional, the problem can be formulated as a convex sparse optimization problem, for which numerous accurate, efficient and robust methods exist. When the subspaces are of high relative dimension (e.g., hyperplanes), the problem is intrinsically non-convex, and existing methods either lack theory, are computationally costly, lack robustness to outliers, or learn hyperplanes one at a time. In this paper, we propose Hyperplane ARangemntent Descent (HARD), a method that robustly learns all the hyperplanes simultaneously by solving a novel non-convex non-smooth $\ell_1$ minimization problem. We provide geometric conditions under which the ground-truth hyperplane arrangement is a coordinate-wise minimizer of our objective. Furthermore, we devise efficient algorithms, and give conditions under which they converge to coordinate-wise minimizes. We provide empirical evidence that HARD surpasses state-of-the-art methods and further show an interesting experiment in clustering deep features on CIFAR-10.

## 1. Introduction

Given a set of data points from a union of *unknown* linear subspaces $\bigcup_{k=1}^{K} \mathcal{H}_k \subset \mathbb{R}^D$, how can we recover the subspaces $\mathcal{H}_k$ as well as determine which points belong to which subspace—in an *unsupervised* fashion? This is the now classical *subspace clustering* problem, which finds numerous applications in computer vision and machine learning; see [1–3] for reviews. The specific case of *hyperplane clustering* (when each $\mathcal{H}_k$ is a hyperplane) has applications in segmenting rigid-body motions [4, 5] and 3D point clouds [6, 7], hybrid system identification [8, 9], mixed linear regression [10, 11], and sparse component analysis [12–14].

When the subspaces are low-dimensional, one can use sparse and low-rank representation techniques to formulate the problem as a convex optimization problem regularized with the $\ell_2$ [15], $\ell_1$ [16, 17] or nuclear norm [18–21], or combinations thereof [22–24], leading to a fruitful line of research with efficient algorithms [25–27] and theoretical guarantees [28–32]. However, for subspaces whose dimension is high relative to $D$, sparsity and low-rankness break down, and so do these methods.

Table 1 summarizes representative hyperplane clustering methods. *RANSAC* [33] is a sampling and consensus algorithm, which finds one hyperplane at a time (not one shot) by randomly sampling points and fitting a hyperplane to them until a good fit is found. It is highly effective in low dimensions, but it is either not robust or suffers from exponential complexity in high dimensions. *K-Hyperplanes* [34, 35] is a $K$-means like

Table 1: Summary of prior work and our contributions.

|  | theory | efficient | robust | one shot |
|---|---|---|---|---|
| RANSAC | little | in low dim. | in low dim. | no |
| $K$-Hyperplanes | little | somewhat | no | **yes** |
| GPCA | **yes** | no | no | **yes** |
| DPCP | **yes** | **yes** | **outliers** | no |
| KH-DPCP | no | somewhat | somewhat | **yes** |
| HARD (ours) | **yes** | **yes** | **outliers** | **yes** |

method tailored to hyperplane clustering. While simple and intuitive, it is inaccurate and not robust to outliers, and it has limited theoretical guarantees (e.g., of convergence to true hyperplanes). *Generalized Principal Component Analysis* (GPCA) [36] is a provably correct algebraic-geometric method that recovers all hyperplanes *in one shot*, but it has exponential complexity and it is not robust to outliers. *Dual Principal Component Pursuit* (DPCP) [37, 38] is a robust hyperplane learning algorithm based on solving a non-convex, non-smooth optimization problem. It is efficient and provably convergent to the true hyperplane, but when

---

*Equal contribution    [1]Mathematical Institute for Data Science, Johns Hopkins University    [2]Center for Innovation in Data Engineering and Science, University of Pennsylvania    Code: https://github.com/tianjiaoding/hard

First Conference on Parsimony and Learning (CPAL 2024).

*directly* extended to hyperplane clustering, DPCP is forced to identify one hyperplane at a time (i.e., not in one shot), which hinders accuracy and theoretical analysis [6, 7]. *KH-DPCP* [6] integrates DPCP into the $K$-Hyperplanes framework. In doing so, it inherits the one-shot ability of $K$-Hyperplanes and the robustness of DPCP to a certain extent. But it also compromises accuracy and comes with no theoretical guarantees.

Then it seems fair to ask: Can we design a hyperplane clustering method that is accurate, efficient, outlier-robust, and supports one-shot recovery with theoretical guarantees?

**Our Contributions.** To design such a method, we blend the GPCA and DPCP approaches in order to explore the best of both worlds. This idea leads us to $\ell_1$ or Huber-style losses for outlier-robust hyperplane-clustering, for which we derive a number of algorithmic, theoretical, and experimental contributions:

• *Algorithms*: We devise algorithms to minimize the proposed $\ell_1$ or Huber-type losses based on the idea of *block coordinate descent* [39]. Since we aim to recover the *hyperplane arrangement* $\bigcup_{k=1}^{K} \mathcal{H}_k$, we name these algorithms HARD-$\ell_1$ and HARD-Huber (*Hyperplane ARrangement Descent*), and thus the title.
• *Theory*: We prove that the ground-truth hyperplane arrangement $\bigcup_{k=1}^{K} \mathcal{H}_k$ is a (*coordinate-wise*) minimizer of our loss function under certain geometric conditions (Theorems 4.2 and 4.3). Moreover, we provide conditions under which HARD-$\ell_1$ and HARD-Huber converge to critical points or (*coordinate-wise*) minimizers (Theorems 5.2 to 5.4). Even though we are not aware of similar results for prior hyperplane clustering methods, our theory does shed light on why the proposed HARD algorithms work well.
• *Experiments*: We show that HARD outperforms state-of-the-art methods on synthetic data with different parameters (e.g., ambient dimension $D$, number of hyperplanes $K$, outlier ratios). We further apply HARD to clustering deep features on the CIFAR-10 dataset [40].

## 2. Prior Art

Prior hyperplane clustering methods are roughly of two types, *outlier-free* [34, 36, 41, 42] and *outlier-robust* [6, 7, 38, 43]. Our review is focused on the two most related methods, GPCA [36] and DPCP [38].

**GPCA.** A linear hyperplane $\mathcal{H}_k$ is identified uniquely up to sign with the unit vector $\boldsymbol{b}_k^*$ lying in the sphere $\mathbb{S}^{D-1} := \{\boldsymbol{y} \in \mathbb{R}^D : \|\boldsymbol{y}\|_2 = 1\}$ and orthogonal to $\mathcal{H}_k$, so $\boldsymbol{x} \in \bigcup_{k=1}^{K} \mathcal{H}_k$ if and only if $\prod_{k=1}^{K}(\boldsymbol{x}^\top \boldsymbol{b}_k^*) = 0$. In other words, a union of $K$ hyperplanes can be represented by a homogeneous polynomial of degree $K$ whose linear factors give the normal vectors to the hyperplanes. Given $N$ data points $\{\boldsymbol{x}_j\}_{j=1}^{N}$ lying in $K$ hyperplanes $\bigcup_{k=1}^{K} \mathcal{H}_k$, instead of solving a large system of polynomial equations

$$\prod_{k=1}^{K}(\boldsymbol{x}_j^\top \boldsymbol{b}_k) = 0, \quad \forall j = 1, \dots, N, \tag{1}$$

GPCA finds all hyperplanes *in one shot* by fitting a single polynomial to the data and factorizing it into linear forms. This observation led to a series of provably correct algorithms for clustering a union of hyperplanes in the absence of noise [36]. However, exact methods suffer from two major disadvantages. First, they exhibit exponential complexity in the dimension of the data $D$ and the number of hyperplanes $K$. Second, they assume noiseless data, as polynomial fitting and factorization methods are known to be sensitive to noise.

To address both challenges, [44] considered a least squares formulation of GPCA, in which one minimizes directly over the normal vectors (thus reducing computational complexity and handling noise)

$$\min_{\{\boldsymbol{b}_k\}_{k=1}^{K}} \sum_{j=1}^{N} \prod_{k=1}^{K}(\boldsymbol{x}_j^\top \boldsymbol{b}_k)^2 \quad \text{s.t.} \quad \boldsymbol{b}_k \in \mathbb{S}^{D-1}, \quad k = 1, \dots, K \tag{GPCA-$\ell_2$}$$

However, at the time, this was considered a difficult problem, not only due to the non-convexity of the objective and constraints, but also due to the existence of permutation symmetries. Indeed, despite recent progress on analyzing problems of this kind, e.g., dictionary learning [45–47] and tensor decomposition [48, 49], we are not aware of a provably convergent method for (GPCA-$\ell_2$). Moreover, the formulation in (GPCA-$\ell_2$) is sensitive to outliers, which we discuss next.

**DPCP.** In practice, the observed data points $\{\tilde{\boldsymbol{x}}_j\}_{j=1}^{M+N} := \{\boldsymbol{x}_j\}_{j=1}^{N} \bigcup \{\boldsymbol{o}_i\}_{i=1}^{M}$ often include a set of $M$ *outlier* points $\{\boldsymbol{o}_i\}_{i=1}^{M}$ *far from* the union $\bigcup_{k=1}^{K} \mathcal{H}_k$ in the (vague) sense that $|\boldsymbol{o}_i^\top \boldsymbol{b}_k^*|$ is *large* for every $k$.

When there is only one hyperplane $\mathcal{H}_1$, *Dual Principal Component Pursuit* (DPCP) [37, 38] finds the normal vector $\boldsymbol{b}_1^*$ by solving the non-convex and non-smooth optimization problem

$$\hat{\boldsymbol{b}}_1 \in \underset{\boldsymbol{b}_1 \in \mathbb{S}^{D-1}}{\operatorname{argmin}} \sum_{j=1}^{M+N} |\tilde{\boldsymbol{x}}_j^\top \boldsymbol{b}_1|. \tag{DPCP}$$

The formulation in (DPCP) dates back at least to the early work of [50], and is related to the *Median $K$-Flats* method (MKF) [43]. It finds a lot of modern applications (e.g., robust subspace recovery [51–55], dictionary learning [56–58], geometric vision [59–61]), and this is perhaps why it has recently received considerable interest [62–67]. However, while existing DPCP algorithms are provably convergent to a global minimum, and (DPCP) is provably correct (the true normal is a global minimum) and highly outlier-robust, existing theory and algorithms can not be applied *directly* to hyperplane clustering, the scenario of our interest.

This motivated [6] to extend (DPCP) to multiple hyperplanes. The first idea of [6] has some *greedy* flavor: solve (DPCP) anyway (even if $K > 1$), as this might yield a $\hat{\boldsymbol{b}}_1$ that is (approximately) orthogonal to most of the points [6, Theorem 6]; then remove or down-weight the points that are (approximately) orthogonal to $\hat{\boldsymbol{b}}_1$; then solve (DPCP) again, but now on the remaining points; and so on. This allows for learning the hyperplane arrangement $\bigcup_{k=1}^K \mathcal{H}_k$ sequentially. This method assumes there is a *dominant hyperplane* [7, 11, 68] containing most of the points; but if without this assumption, $\hat{\boldsymbol{b}}_1$ might be inaccurate and damages the subsequent estimation of hyperplanes. The second idea of [6] is to integrate (DPCP) into the above-mentioned KH framework [34], i.e., update the hyperplanes via (DPCP) instead of *Principal Component Analysis* (PCA) (cf. §C.1). While such integration improves accuracy, it inevitably inherits two drawbacks from KH. First is the sensitivity to initialization, which can be partially alleviated by multiple initializations [7]. Second, the underlying theory of why such an integration works well has thus far remained, to our knowledge, obscure.

## 3. Overview: Formulation and Algorithms

In this section, we introduce novel objective functions for hyperplane clustering in the presence of outliers (§3.1) and provide practical algorithms for minimizing them (§3.2). The emphasis of this section is on giving intuition and implementation recipes, while theoretical insights of our development are left to §4 and §5.

### 3.1. Problem Setup and Formulation

**Problem Setup.** We collect the notations used previously to set up the problem. Let $\bigcup_{k=1}^K \mathcal{H}_k$ be a union of hyperplanes of $\mathbb{R}^D$, and denote by $\boldsymbol{b}_k^*$ the (unique up to sign) normal vector of $\mathcal{H}_k$. Let $\{\boldsymbol{x}_j\}_{j=1}^N$ be a set of $N$ points lying in $\bigcup_{k=1}^K \mathcal{H}_k$; call each $\boldsymbol{x}_j$ an *inlier*. Let $\{\boldsymbol{o}_i\}_{i=1}^M$ be a set of $M$ points of $\mathbb{R}^D$ *far from the union* $\bigcup_{k=1}^K \mathcal{H}_k$; call each $\boldsymbol{o}_i$ an *outlier*. Given the set of data points $\{\tilde{\boldsymbol{x}}_j\}_{j=1}^{M+N} := \{\boldsymbol{x}_j\}_{j=1}^N \bigcup \{\boldsymbol{o}_i\}_{i=1}^M$, we aim to recover all hyperplanes, cluster all inliers into these hyperplanes, as well as identify and remove all outliers.

**Formulation.** Although we do not know whether each data point $\tilde{\boldsymbol{x}}_j$ is an inlier or outlier, we do know that, if $\tilde{\boldsymbol{x}}_j$ is indeed an inlier, then the product $\prod_{k=1}^K |\tilde{\boldsymbol{x}}_j^\top \boldsymbol{b}_k^*|$ is equal to $0$, or otherwise this product would be *large*. Therefore, to gain robustness to outliers, we propose to solve

$$\min_{\{\boldsymbol{b}_k\}_{k=1}^K} \sum_{j=1}^{N+M} \prod_{k=1}^K |\tilde{\boldsymbol{x}}_j^\top \boldsymbol{b}_k| \quad \text{s.t.} \quad \boldsymbol{b}_k \in \mathbb{S}^{D-1}, \quad k = 1, \dots, K \tag{GPCA-$\ell_1$}$$

(GPCA-$\ell_1$) has an $\ell_1$-style objective and is an unsquared version of (GPCA-$\ell_2$); if $\prod_{k=1}^K |\tilde{\boldsymbol{x}}_j^\top \boldsymbol{b}_k|$ were the residual term for linear regression, then (GPCA-$\ell_1$) would become *least absolute deviation*, a classic problem that is known to be outlier-robust [69]. We will soon see the robustness of (GPCA-$\ell_1$) to outliers as well.

To finish the section, we make some remarks and teasers:

• (*Invariance*) (GPCA-$\ell_1$) is *sign-invariant*, i.e., changing the sign of $\boldsymbol{b}_k$ has no effect whatsoever on the objective value. Also, (GPCA-$\ell_1$) is *permutation-invariant*, e.g., swapping $\boldsymbol{b}_1$ and $\boldsymbol{b}_2$ does not alter the value of the product $|\tilde{\boldsymbol{x}}_j^\top \boldsymbol{b}_1| \cdot |\tilde{\boldsymbol{x}}_j^\top \boldsymbol{b}_2|$ or the objective value.

- (*Harder GPCA?*) While (GPCA-$\ell_1$) can be regarded as robustifying (1) and (GPCA-$\ell_2$) to outliers, the cost of such robustification is the need to solve a *non-smooth polynomial optimization problem* (GPCA-$\ell_1$), which appears harder (at least conceptually) than solving equations (1) or the smooth problem (GPCA-$\ell_2$). Moreover, to our knowledge, there are no *polynomial-time* algorithms proposed for (1) and no working algorithms ever proposed for (GPCA-$\ell_2$)—How can we solve the non-smooth problem (GPCA-$\ell_1$) efficiently?

- (*Harder DPCP?*) In the special case of a single hyperplane, (GPCA-$\ell_1$) is exactly the same as (DPCP). Recent theoretical advances on this subject seem to give way to the open problem of proving a given algorithm for (DPCP) is globally convergent (cf. [70, Table II]). Under such circumstances, how can we devise provably and globally convergent methods for the harder problem (GPCA-$\ell_1$)?

- (*Correctness*) Do the true normal vectors form a global minimizer of (GPCA-$\ell_1$)? This is certainly the case when there are no outliers, in which case the optimal value is zero. Critical to the theory behind (DPCP) is to identify conditions on the data under which the global minimum is still the true normal vector despite the $M$ extra terms in the objective. Controlling these extra terms should be clearly harder for (GPCA-$\ell_1$).

## 3.2. Hyperplane Arrangement Descent Algorithms

This section offers intuition and practical recipes for solving (GPCA-$\ell_1$); theoretical analysis is deferred to §4 and §5. The proposed family of methods is termed "HARD" (*hyperplane arrangement descent*); cf. Figure 1.

### 3.2.1. HARD-L1 and HARD-L1+

**HARD-$\ell_1$.** It seems hard to solve (GPCA-$\ell_1$) as it has a non-convex and non-smooth objective with non-convex constraints. However, the objective is *marginally convex*, e.g., it is convex in $\boldsymbol{b}_1$ (with $\boldsymbol{b}_2, \ldots, \boldsymbol{b}_K$ fixed). This suggests minimizing (GPCA-$\ell_1$) in a *coordinate-wise* manner, resulting in what we call the "HARD-$\ell_1$" algorithm (Algorithm 1). Specifically, HARD-$\ell_1$ initializes at $\{\boldsymbol{b}_k^{(0)}\}_{k=1}^K$, and it computes $\boldsymbol{b}_k^{(t+1)}$ at iteration $t$ for every $k$, by minimizing a weighted $\ell_1$ objective over the sphere $\mathbb{S}^{D-1}$ ($\ell_1$). Weight $w_{j,k}^{(t)}$ is calculated such that ($\ell_1$) corresponds exactly to (GPCA-$\ell_1$) with all other normal vectors given except $\boldsymbol{b}_k$.

The core step of HARD-$\ell_1$ is solving ($\ell_1$). This is a weighted version of (DPCP) [37, 38], to which many methods [38, 62–65] can be applied. However, these DPCP algorithms are *iterative*, so integrating them into HARD-$\ell_1$ would require some criteria to terminate (controlled by some hyperparameters) as well as multiple runs for just a single update of $\boldsymbol{b}_k^{(t)}$. This is why applying these methods to ($\ell_1$) might make HARD-$\ell_1$ cumbersome and inefficient.

---

**Algorithm 1:** HARD-$\ell_1$

Initialization: $\{\boldsymbol{b}_k^{(0)}\}_{k=1}^K \subset \mathbb{S}^{D-1}$

For $t \leftarrow 0, 1, \ldots$:

For $k \leftarrow 1, \ldots, K$:

$$w_{j,k}^{(t)} \leftarrow \begin{cases} \prod_{i \neq k} |\tilde{\boldsymbol{x}}_j^\top \boldsymbol{b}_i^{(t)}| & k = 1 \\ \prod_{i < k} |\tilde{\boldsymbol{x}}_j^\top \boldsymbol{b}_i^{(t+1)}| \cdot \prod_{i > k} |\tilde{\boldsymbol{x}}_j^\top \boldsymbol{b}_i^{(t)}| & k > 1 \end{cases}$$

$$\boldsymbol{b}_k^{(t+1)} \in \underset{\boldsymbol{b}_k \in \mathbb{S}^{D-1}}{\operatorname{argmin}} \sum_{j=1}^{N+M} w_{j,k}^{(t)} \cdot |\tilde{\boldsymbol{x}}_j^\top \boldsymbol{b}_k| \qquad (\ell_1)$$

---

**HARD-$\ell_1$+.** These considerations bring us to our next insight, the idea of replacing ($\ell_1$) by

$$\boldsymbol{b}_k^{(t+1)} \in \underset{\boldsymbol{b}_k \in \mathbb{S}^{D-1}}{\operatorname{argmin}} \sum_{j=1}^{N+M} w_{j,k}^{(t)} \cdot \frac{(\tilde{\boldsymbol{x}}_j^\top \boldsymbol{b}_k)^2}{\max\{|\tilde{\boldsymbol{x}}_j^\top \boldsymbol{b}_k^{(t)}|, \delta\}}, \qquad (\ell_1+)$$

leading to what we call the HARD-$\ell_1$+ method. In ($\ell_1$+), $\delta$ is a small positive number to avoid division by zero. A crucial advantage of ($\ell_1$+) over ($\ell_1$) is that ($\ell_1$+) can be solved via SVD. Moreover, note that

$$\frac{(\tilde{\boldsymbol{x}}_j^\top \boldsymbol{b}_k)^2}{\max\{|\tilde{\boldsymbol{x}}_j^\top \boldsymbol{b}_k^{(t)}|, \delta\}} + \max\{|\tilde{\boldsymbol{x}}_j^\top \boldsymbol{b}_k^{(t)}|, \delta\} \geq 2|\tilde{\boldsymbol{x}}_j^\top \boldsymbol{b}_k|,$$

thus ($\ell_1$+) is always an upper bound of ($\ell_1$) (plus some constant terms). Consequently, minimizing such upper bound ($\ell_1$+) would also decrease the objective values of ($\ell_1$). This is related to the so-called *majorization minimization* paradigm [71], and offers a loose explanation as to why HARD-$\ell_1$+ is a reasonable algorithm.

### 3.2.2. HARD-Huber and HARD-Huber+

**HARD-Huber.** While HARD-$\ell_1$+ only requires an SVD to solve ($\ell_1$+) and thus can be implemented very efficiently, the objective (GPCA-$\ell_1$) is still non-smooth, which creates obstacles for convergence analysis. This motivates us to consider a Huber-like loss $h(\cdot)$ to smooth the $\ell_1$ norm:

$$h(r) = |r| \quad \text{for} \quad |r| > \delta, \quad \text{or} \quad \frac{r^2 + \delta^2}{2\delta} \quad \text{for} \quad |r| \le \delta.$$

Here, $\delta$ is some positive and small constant; see also ($\ell_1$+). Indeed, $h(r)$ is exactly equal to $|r|$ except when $|r|$ is small, in which case $h(r)$ is quadratic and in fact $h(r) \to |r|$ as $\delta \to 0$. Notably, $h(r)$ is smooth, differently from $|r|$. With this Huber-like loss $h(r)$, we now revise (GPCA-$\ell_1$) into

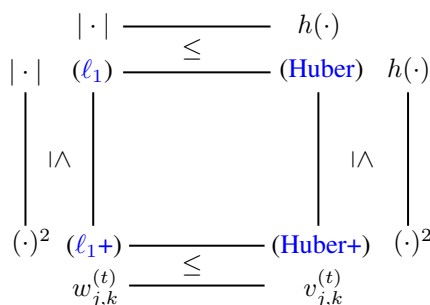

$$\min_{\{\boldsymbol{b}_k\}_{k=1}^K} \sum_{j=1}^{N+M} \prod_{k=1}^K h(\tilde{\boldsymbol{x}}_j^\top \boldsymbol{b}_k) \qquad \text{(GPCA-Huber)}$$

$$\text{s.t.} \quad \boldsymbol{b}_k \in \mathbb{S}^{D-1}, \quad k = 1, \ldots, K$$

and revise HARD-$\ell_1$ into the *HARD-Huber* algorithm:

$$v_{j,k}^{(t)} \leftarrow \begin{cases} \prod_{i \ne k} h(\tilde{\boldsymbol{x}}_j^\top \boldsymbol{b}_i^{(t)}) & k = 1 \\ \prod_{i<k} h(\tilde{\boldsymbol{x}}_j^\top \boldsymbol{b}_i^{(t+1)}) \cdot \prod_{i>k} h(\tilde{\boldsymbol{x}}_j^\top \boldsymbol{b}_i^{(t)}) & k > 1 \end{cases}$$

$$\boldsymbol{b}_k^{(t+1)} \in \operatorname*{argmin}_{\boldsymbol{b}_k \in \mathbb{S}^{D-1}} \sum_{j=1}^{N+M} v_{j,k}^{(t)} \cdot h(\tilde{\boldsymbol{x}}_j^\top \boldsymbol{b}_k) \qquad \text{(Huber)}$$

HARD-Huber differs from HARD-$\ell_1$ in that it replaces the absolute value $|\tilde{\boldsymbol{x}}_j^\top \boldsymbol{b}_k|$ into $h(\tilde{\boldsymbol{x}}_j^\top \boldsymbol{b}_k)$; this offers theoretical advantages as we will show in §5.

Figure 1: Relations of HARD methods (§3.2): since $|\cdot| \le h(\cdot)$, (Huber) upper bounds ($\ell_1$); since $|\cdot| \le (\cdot)^2$, ($\ell_1$+) upper bounds ($\ell_1$); since $h(\cdot) \le (\cdot)^2$, (Huber+) upper bounds (Huber); since $w_{j,k}^{(t)} \le v_{j,k}^{(t)}$, (Huber+) upper bounds ($\ell_1$+).

**HARD-Huber+.** Similarly to ($\ell_1$), solving each subproblem (Huber) iteratively is undesired. Similarly to ($\ell_1$+), the practical recipe we advocate is replacing (Huber) with

$$\boldsymbol{b}_k^{(t+1)} \in \operatorname*{argmin}_{\boldsymbol{b}_k \in \mathbb{S}^{D-1}} \sum_{j=1}^{N+M} v_{j,k}^{(t)} \cdot \frac{(\tilde{\boldsymbol{x}}_j^\top \boldsymbol{b}_k)^2}{\max\{|\tilde{\boldsymbol{x}}_j^\top \boldsymbol{b}_k^{(t)}|, \delta\}} \qquad \text{(Huber+)}$$

which gives the *HARD-Huber+* method. It is now clear that (Huber+) is to (Huber) what ($\ell_1$+) is to ($\ell_1$) and the difference of ($\ell_1$+) and (Huber+) is in the weights, $w_{j,k}^{(t)}$ and $v_{j,k}^{(t)}$; see also Figure 1. We will mainly use HARD-$\ell_1$+ and HARD-Huber+ for practical purposes. That said, the convergence properties of these HARD algorithms, as well as their intriguing interplay, are of sufficient theoretical interests, and will be explored in greater detail in §5.

## 4. Analysis of the Objective

In this section, we first define some geometric quantities that depend on the distribution of inliers, outliers and ground-truth normal vectors $\{\pm\boldsymbol{b}_k^*\}_{k=1}^K$ (§4.1). Then, we show that $\{\pm\boldsymbol{b}_k^*\}_{k=1}^K$ are coordinate-wise minimizers (Definition 4.1) of (GPCA-$\ell_1$), under conditions that depend on these geometric quantities (§4.2).

### 4.1. Geometric Quantities

Let $\{\boldsymbol{x}_i^k\}_{i=1}^{N_k}$ be the $N_k$ inliers from $\mathcal{H}_k$, and note that $N = N_1 + \cdots + N_K$. Define $d_j^k := \prod_{i \ne k}^K |\boldsymbol{x}_j^{k\top} \boldsymbol{b}_i^*| \ge 0$ which measures how the $j$-th inlier $\boldsymbol{x}_j^k$ of $\mathcal{H}_k$ is close to other $K-1$ hyperplanes. Similarly, let $q_j^k := \prod_{i \ne k}^K |\boldsymbol{o}_j^\top \boldsymbol{b}_i^*|$ measure how the $j$-th outlier $\boldsymbol{o}_j$ is close to all $\mathcal{H}_i$'s such that $i \ne k$. With these, we can now define the following

$$c_{\text{in},k,\min} := \frac{1}{N_k} \min_{\boldsymbol{b} \in \mathcal{H}_k \cap \mathbb{S}^{D-1}} \sum_{j=1}^{N_k} d_j^k \cdot |\boldsymbol{x}_j^{k\top} \boldsymbol{b}| \tag{2}$$

$$c_{\text{out},k} := \frac{1}{M}\left(\max_{\boldsymbol{b}\in\mathbb{S}^{D-1}}\sum_{j=1}^{M}q_j^k\cdot\left|\boldsymbol{o}_j^\top\boldsymbol{b}\right| - \min_{\boldsymbol{b}\in\mathbb{S}^{D-1}}\sum_{j=1}^{M}q_j^k\cdot\left|\boldsymbol{o}_j^\top\boldsymbol{b}\right|\right)$$

$$\bar{\eta}_{\text{out},k} := \frac{1}{M}\max_{\boldsymbol{b}\in\mathbb{S}^{D-1}}\left\|(\boldsymbol{I}-\boldsymbol{b}\boldsymbol{b}^\top)\sum_{j=1}^{M}q_j^k\,\text{sign}(\boldsymbol{o}_j^\top\boldsymbol{b})\boldsymbol{o}_j\right\|_2 + \frac{D}{M}.$$

Intuitively, these quantities measure the uniformity of inliers within each hyperplane, that of outliers, as well as how inliers from one hyperplane are far from other hyperplanes. For example, $c_{\text{in},k,\min}$ is large if there are sufficiently many inliers of $\mathcal{H}_k$ that are far from other hyperplanes $\mathcal{H}_i(i\neq k)$ and uniformly distributed in $\mathcal{H}_k$. Similarly, if outliers far from all hyperplanes are sufficiently uniformly distributed in $\mathbb{S}^{D-1}$, then $c_{\text{out},k}$ and $\bar{\eta}_{\text{out},k}$ are small. These geometric quantities turn out to be useful in our results below.

## 4.2. Coordinate-Wise Minimizers

We first formally define coordinate-wise minimizers:

**Definition 4.1.** A feasible point $\{\bar{\boldsymbol{b}}_k\in\mathbb{S}^{D-1}\}_{k=1}^K$ of a program $\min_{\{\boldsymbol{b}_i\in\mathbb{S}^{D-1}\}_{i=1}^K}f(\boldsymbol{b}_1,\ldots,\boldsymbol{b}_K)$ is said to be a coordinate-wise minimizer of this program, if it holds for all $\boldsymbol{b}_k\in\mathbb{S}^{D-1}$ and $k=1,\ldots,K$ that

$$f(\bar{\boldsymbol{b}}_1,\ldots,\bar{\boldsymbol{b}}_K)\leq f(\bar{\boldsymbol{b}}_1,\ldots,\bar{\boldsymbol{b}}_{k-1},\boldsymbol{b}_k,\bar{\boldsymbol{b}}_{k+1},\ldots,\bar{\boldsymbol{b}}_K).$$

With this definition, we will show that under conditions on the geometric quantities (2), the ground-truth normal vectors $\{\pm\boldsymbol{b}_k^*\}_{k=1}^K$ are coordinate-wise minimizers of (GPCA-$\ell_1$). In fact, we will prove that:

$$\{\pm\boldsymbol{b}_k^*\} = \operatorname*{argmin}_{\boldsymbol{b}_k\in\mathbb{S}^{D-1}}\sum_{j=1}^{N+M}|\tilde{\boldsymbol{x}}_j^\top\boldsymbol{b}_k|\prod_{i\neq k}|\tilde{\boldsymbol{x}}_j^\top\boldsymbol{b}_i^*|, \tag{3}$$

for $k=1,\ldots,K$, which is a slightly stronger claim. Now, we are ready to state our first result:

**Theorem 4.2.** *Any critical point of* (3) *either equals* $\pm\boldsymbol{b}_k^*$, *or has a principal angle from* $\mathcal{H}_k$ *smaller than or equal to* $\arcsin\left(\frac{M\bar{\eta}_{\text{out},k}}{N_k c_{\text{in},k,\min}}\right)$.

Theorem 4.2 says that any critical point of (3) is either normal vectors $\{\pm\boldsymbol{b}_k^*\}$ to $\mathcal{H}_k$, or must be sufficiently far away from them. Thus, if one can show that the objective of (3) is larger in the latter case than the former, then $\{\pm\boldsymbol{b}_k^*\}$ must be the solution to (3). This leads to:

**Theorem 4.3.** *Any minimizer of* (3) *must be* $\pm\boldsymbol{b}_k^*$ *if*

$$\frac{M}{N_k}\frac{\sqrt{c_{\text{out},k}^2+\bar{\eta}_{\text{out},k}^2}}{c_{\text{in},k,\min}} < 1. \tag{4}$$

*Further, if* (4) *holds for* $k=1,\ldots,K$, *then* $\{\pm\boldsymbol{b}_k^*\}_{k=1}^K$ *are coordinate-wise minimizers of* (GPCA-$\ell_1$).

We give some intuitions on Theorem 4.3. Firstly, for (4) to hold, we need the outlier ratio $M/N_k$ with respect to hyperplane $\mathcal{H}_k$ to be small, and we require inliers of $\mathcal{H}_k$ and outliers to be well distributed (§4.1) so that $c_{\text{in},k,\min}$ is large and $c_{\text{out},k},\bar{\eta}_{\text{out},k}$ are small. Note further that (4) decouples among data from different hyperplanes, in that for each $k$ (4) does not depend on inliers of hyperplanes other than $\mathcal{H}_k$.

Theorems 4.2 and 4.3 are motivated by [62] and [72]. However, [62] concerns only a single hyperplane and [72, Lemma 18] a single dominant hyperplane. Theorem 4.3 is an extension to the scenario of multiple hyperplanes and recovers the result of [62] when $K=1$.

# 5. Convergence of the HARD Algorithms

In §4 we provided conditions guaranteeing $\{\pm\boldsymbol{b}_k^*\}_{k=1}^K$ are coordinate-wise minimizers and coordinate-wise critical points of (GPCA-$\ell_1$). Here, we will prove that our HARD algorithms converge to coordinate-wise minimizers or critical points. We start with the following assumption:

**Assumption 5.1** (Unique Block Minimizer). For any $k = 1, \ldots, K$, any $\boldsymbol{b}_i$'s ($i \neq k$), and any $\boldsymbol{b}'_k \in \mathbb{S}^{D-1}$, each of the following has a unique (up to sign) global minimizer:

$$\min_{\boldsymbol{b}_k \in \mathbb{S}^{D-1}} \sum_{j=1}^{N+M} |\tilde{\boldsymbol{x}}_j^\top \boldsymbol{b}_k| \cdot \prod_{i \neq k} |\tilde{\boldsymbol{x}}_j^\top \boldsymbol{b}_i| \tag{A1}$$

$$\min_{\boldsymbol{b}_k \in \mathbb{S}^{D-1}} \sum_{j=1}^{N+M} h(\tilde{\boldsymbol{x}}_j^\top \boldsymbol{b}_k) \cdot \prod_{i \neq k} h(\tilde{\boldsymbol{x}}_j^\top \boldsymbol{b}_i) \tag{A2}$$

$$\min_{\boldsymbol{b}_k \in \mathbb{S}^{D-1}} \sum_{j=1}^{N+M} \frac{(\tilde{\boldsymbol{x}}_j^\top \boldsymbol{b}_k)^2}{\max\{|\tilde{\boldsymbol{x}}_j^\top \boldsymbol{b}'_k|, \delta\}} \cdot \prod_{i \neq k} h(\tilde{\boldsymbol{x}}_j^\top \boldsymbol{b}_i) \tag{A3}$$

Assumption 5.1 assumes the uniqueness of the global minimizer when minimizing (A1), (A2), or (A3) with respect to each block of variables $\boldsymbol{b}_k$. Such assumption is very standard and in some sense *necessary* for proving convergence of *block coordinate descent* methods [39, 73]. We now provide some intuitive justification for Assumption 5.1 in our environment. Note that [38, Theorem 5] proves the *continuous version* of (DPCP) always has a global minimizer $\pm \boldsymbol{b}_k^*$, which sheds light on the weighted DPCP problem (A1). Intuitively (but not necessarily rigorously), (A2) is more likely than (A1) to have a unique global minimizer as $h$ is further locally quadratic (strongly convex). We emphasize that, still, (A1) and (A2) are hard to verify and might be considered as strong assumptions. On the other hand, (A3) has a unique global minimizer as long as the least two eigenvalues of the weighted covariance matrix $\sum_{j=1}^{N+M} c_j \cdot \tilde{\boldsymbol{x}}_j \tilde{\boldsymbol{x}}_j^\top$ are distinct ($c_j := \prod_{i \neq k} h(\tilde{\boldsymbol{x}}_j^\top \boldsymbol{b}_i) / \max\{|\tilde{\boldsymbol{x}}_j^\top \boldsymbol{b}'_k|, \delta\}$), and this is expected to be true given enough (random) data points; in fact, this is always true in our experiments where we checked that numerically the difference between the least two eigenvalues is typically large.

In what follows, we will introduce three global convergence results, Theorems 5.2 to 5.4. Theorem 5.2 can be proved by only assuming that (A1) has a unique global minimizer ($\forall k, \forall \boldsymbol{b}_i$'s, $\forall \boldsymbol{b}'_k$); Theorem 5.3, assuming (A2); Theorem 5.4, assuming (A3). However, we will use Assumption 5.1 consistently for simplicity.

## 5.1. HARD-L1

We now characterize the convergence of HARD-$\ell_1$:

**Theorem 5.2.** *Under Assumption 5.1 (A1), any limit point of the sequence $\{(\boldsymbol{b}_1^{(t)}, \ldots, \boldsymbol{b}_K^{(t)})\}_t$ generated by the HARD-$\ell_1$ algorithm is a coordinate-wise minimizer of (GPCA-$\ell_1$).*

The proof of Theorem 5.2 is motivated by Proposition 2.7.1 of [39]. In fact, the latter requires the constraint sets to be *convex*, and we extend it for the *non-convex* spherical constraint $\mathbb{S}^{D-1}$ to obtain Theorem 5.2.

Can we then also prove a similar result to Theorem 5.2 for HARD-$\ell_1$+? The answer that we have at this moment is, unfortunately, no. The difficulty of proving so is that HARD-$\ell_1$+ only minimizes an upper bound ($\ell_1$+) of ($\ell_1$), not exactly ($\ell_1$). However, we will be able to prove convergence properties of HARD-Huber+ in §5.2. Moreover, $h(r)$ upper bounds $|r|$, thus (Huber+) upper bounds ($\ell_1$+); recall Figure 1 and §3.2. Therefore the convergence of HARD-Huber+ would shed light on the behavior of HARD-$\ell_1$+.

## 5.2. HARD-Huber

**HARD-Huber.** While §5.1 derived a basic guarantee for HARD-$\ell_1$ to converge to coordinate-wise minimizers (Theorem 5.2), the non-smooth nature of the $\ell_1$ loss of (GPCA-$\ell_1$) makes it hard to establishing convergence guarantees to critical points. On the other hand, since (GPCA-Huber) has a smooth objective, we are able to derive such guarantees for the HARD-Huber algorithm:

**Theorem 5.3.** *Under Assumption 5.1 (A2), any limit point of the iterates $\{(\boldsymbol{b}_1^{(t)}, \ldots, \boldsymbol{b}_K^{(t)})\}_t$ generated by the HARD-Huber algorithm is a coordinate-wise minimizer and also a critical point of (GPCA-Huber).*

Convergence of HARD-Huber to coordinate-wise minimizers in Theorem 5.3 is of a similar flavor to Theorem 5.2. An extra result in Theorem 5.3, though, is that HARD-Huber also converges to critical points, and this is made possible by leveraging the smoothness of (GPCA-Huber).

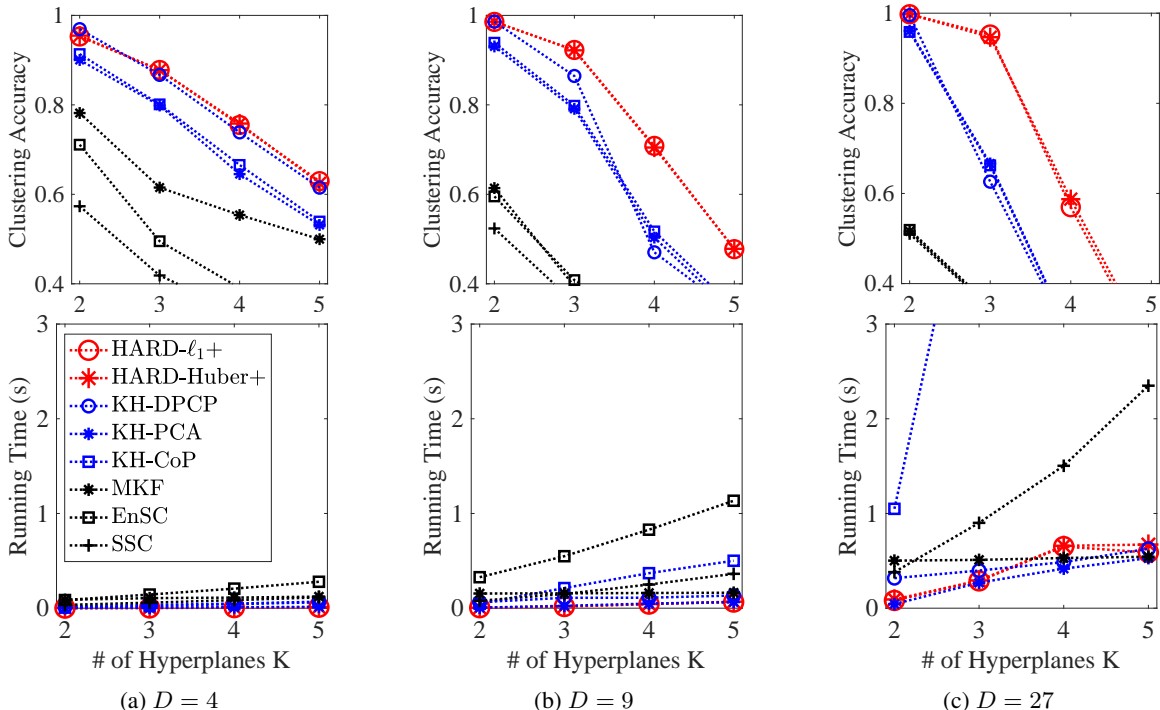

Figure 2: Clustering accuracy and running time of different methods on synthetic union-of-hyperplane data (§6.1) over 100 repeated experiments. With $50(D-1)$ inliers from each hyperplane and $30\%$ outliers, the proposed HARD-$\{\ell_1+,$Huber+$\}$ are the most accurate and fastest.

**HARD-Huber+.** Since HARD-Huber+ is easy to implement in practice, we arm it with a convergence result:

**Theorem 5.4.** *Under Assumption 5.1 (A3), any limit point of the iterates $\{(\boldsymbol{b}_1^{(t)}, \dots, \boldsymbol{b}_K^{(t)})\}_t$ generated by the HARD-Huber+ algorithm is a critical point of* (GPCA-Huber).

Theorem 5.4 is interesting as it shows that HARD-Huber+ converges to critical points of (GPCA-Huber), even if (Huber+) differs from (Huber). This difference, however, is double-edged: It is harder, if not impossible, to prove HARD-Huber+ converges to coordinate-wise minimizers.

The proof device that makes Theorem 5.4 available is quite similar to that of [74], and the shared principle is *block successive upper-bound minimization*: (Huber+) is a carefully constructed upper bound of (Huber) (plus some constant) that makes use of the smoothness of (GPCA-Huber). Unfortunately, that construction does not apply to $(\ell_1+)$ as (GPCA-$\ell_1$) is not smooth. Finally, we note a key difference from [74]: We have non-convex constraints $\boldsymbol{b}_k \in \mathbb{S}^{D-1}$, while [74] makes convexity assumptions on constraints.

# 6. Experiments

## 6.1. Clustering Synthetic Data

**Data.** We generate $K$ hyperplanes $\{\mathcal{H}_k\}_{k=1}^K$ in $\mathbb{R}^D$ uniformly at random. For each $k = 1, \dots, K$, we sample $N_k = 50(D-1)$ points from $\mathrm{Unif}(\mathcal{H}_k \cap \mathbb{S}^{D-1})$, totalling $N = 50K(D-1)$ inliers. We sample $M$ outliers from $\mathrm{Unif}(\mathbb{S}^{D-1})$ to have a certain outlier ratio $M/(N + M)$.

**Metrics.** We run the algorithms to estimate a hyperplane arrangement $\bigcup_{k=1}^K \hat{\mathcal{H}}_k$ with $K$ given. Assigning each point to its closest estimated hyperplane, we get the clusters. This allows us to report *clustering accuracy*, which is the number of correctly clustered inliers divided by $N$. We further assess how well the algorithms separate outliers from inliers. By comparing the distance of each point to $\bigcup_{k=1}^K \hat{\mathcal{H}}_k$ to a threshold we obtain an estimated inlier set, which in turn gives precision and recall. Enumerating all possible thresh-

olds, we report the *area under the precision-recall curve* (AUC-ROC). We further report the *F1-score* using a specific threshold $10^{-2}$ on such distance. Finally, we report the *running time*[2] of each algorithm in seconds.

**Methods.** We compare our HARD algorithms with state-of-the-art subspace clustering methods, including Median K-Flats (MKF) [43], Elastic-net Subspace Clustering (EnSC) with active-set solver [24], and Sparse Subspace Clustering (SSC) with orthogonal matching pursuit solver [25]. We further compare with K-Hyperplane (§2) variants, which use PCA (KH-PCA), Coherence Pursuit (KH-CoP) [75] and (DPCP) (KH-DPCP) [7] to estimate a hyperplane for each cluster. Note that the latter two have been shown to achieve state-of-the-art performance in robust subspace and hyperplane clustering respectively. We run all methods with default or otherwise appropriate parameters.

**Fixed $M/(N+M)$, varying $K, D$.** We first fix an outlier ratio of $M/(N+M) = 30\%$, and vary the number of hyperplanes $K \in \{2, 3, 4, 5\}$ and ambient dimension $D \in \{4, 9, 27\}$. Figure 2 shows clustering accuracy and running time over 100 trials. To begin with, with more hyperplanes the problem becomes harder, and all methods yield lower accuracy. Remarkably, HARD is the most accurate in all settings of $K, D$, as seen by the red curves in the left column surpassing other curves. In particular, the performance gap between HARD and KH-DPCP gets larger in higher dimensions $D$, e.g., for $K = 3$, the gap increases from $1.1\%$ ($D = 4$) to $32.8\%$ ($D = 27$). Finally, MKF, EnSC, SSC do not perform well as expected (§1), therefore they are excluded from the experiments henceforth. Limited by space, we leave more experiments in Appendix B.

### 6.2. Clustering Deep Features on CIFAR-10

Finally, we perform an interesting experiment showing the potential of using hyperplane clustering for deep features of natural images. Inspired by the success of subspace clustering, numerous works have proposed to use a *feature map* that sends natural images to a union of subspaces. Such a map can be hand-crafted [76], or learned by using a neural network and applying subspace clustering losses on the image of data points or *features* [77–85].

We take the features learned by an unsupervised method [85] on CIFAR-10 [40], which are $5 \cdot 10^4$ points in $\mathbb{S}^{128}$. We apply a random orthogonal projection on them to points of $\mathbb{R}^{24}$, followed by normalizing them to $\mathbb{S}^{23}$. We run different methods on the processed data to estimate 10 hyperplanes, and report in Figure 3 clustering accuracy over 20 repeated experiments. As seen, HARD is considerably more accurate than KH-DPCP. In particular, HARD-Huber+ has median and max clustering accuracy of $73\%$ and $81\%$. Encouraging as it may sound, [85] and the aforementioned works essentially promote the features to lie on *low-dimensional* subspaces (see, e.g., [83, The-

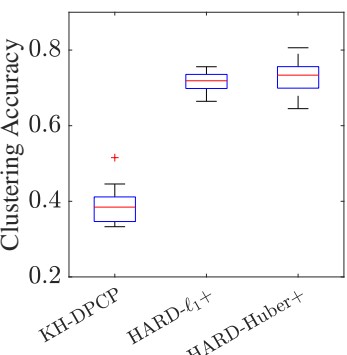

Figure 3: Clustering accuracy of different methods on randomly projected deep features in $\mathbb{R}^{24}$ extracted from $5 \cdot 10^4$ images from CIFAR-10, over 20 repeated experiments.

orem 2.1]), which is why a random projection on data is necessary before hyperplane clustering. Therefore, an interesting subject of research would be to learn feature maps such that the images of data points lie on hyperplane arrangements.

## 7. Conclusion and Limitations

In this paper, we considered the hyperplane clustering problem in the presence of outliers. We showed that the proposed algorithms (HARD) are efficient, outlier-robust, allow for one-shot recovery with theoretical guarantees, and establish state-of-the-art performance. However, our experiments showed that the accuracy of HARD drops as the number of hyperplanes increases, and clustering more than 5 hyperplanes with greater accuracy is still up to future research. From a theoretical viewpoint, our theorems guarantee the convergence of HARD to critical points or (coordinate-wise) minimizers, and future work would consist of proving stronger results that guarantee convergence to the true hyperplane arrangements.

---

[2] The experiments are conducted on a MacBook Pro with M2 Pro chip and 32GB memory.

# Acknowledgements

This work was partially supported by the Northrop Grumman Mission Systems Research in Applications for Learning Machines (REALM) initiative, DARPA Grant HR00112020010, NSF grant 1704458, and ONR MURI 503405-78051. The first author is thankful to Dr. Manolis C. Tsakiris for excellent insights and discussions that motivated the initial development of this project.

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

## A. Proofs

### A.1. Objective Analysis

This section gives the proofs for Theorem 4.2 and Theorem 4.3. We first need a light assumption, stated here:

**Assumption A.1** (Outliers are in general position). *Any subset of $D$ elements from $\{o_i\}_{i=1}^M$ are linearly independent.*

Indeed, with random $\{o_i\}_{i=1}^M$ this assumption is satisfied with probability one. Since both theorems concerns (3), we share an insight here that the objective of (3) is equivalent to

$$g_k(\boldsymbol{b}) := \sum_{j=1}^{N_k} d_j^k |\boldsymbol{x}_j^{k\top} \boldsymbol{b}| + \sum_{j=1}^{M} q_j^k |\boldsymbol{o}_j^\top \boldsymbol{b}|. \tag{5}$$

$$\left( = \sum_{j=1}^{N_k} |\boldsymbol{x}_j^{k\top} \boldsymbol{b}| \prod_{i \neq k}^{K} |\boldsymbol{x}_j^{k\top} \boldsymbol{b}_i^*| + \sum_{j=1}^{M} |\boldsymbol{o}_j^\top \boldsymbol{b}| \prod_{i \neq k}^{K} |\boldsymbol{o}_j^\top \boldsymbol{b}_i^*| \right)$$

Notably, this objective does not involve (the distribution of) inliers $\{\boldsymbol{x}_i^k\}_{i=1}^{N_k} (i \neq k)$ to hyperplanes other than $\mathcal{H}_k$. Now we recall theorem 4.2 and give its proof here.

**Theorem 4.2.** *Any critical point of (3) either equals $\pm\boldsymbol{b}_k^*$, or has a principal angle from $\mathcal{H}_k$ smaller than or equal to $\arcsin\left(\frac{M\bar{\eta}_{\mathrm{out},k}}{N_k c_{\mathrm{in},k,\min}}\right)$.*

*Proof.* Let $\boldsymbol{b}$ be a critical point of (3) that is not orthogonal to $\mathcal{H}_k$, i.e., $\boldsymbol{b} \neq \pm\boldsymbol{b}_k^*$.

**Size of the Riemannian subgradient, inliers only.** Define the subdifferential of $|\cdot|$ as

$$\mathrm{Sgn}(x) = \begin{cases} 1 & x > 0 \\ -1 & x < 0 \\ [-1,1] & x = 0 \end{cases}. \tag{6}$$

With some abuse of notation let $\mathrm{sgn}(x)$ denote any element in $\mathrm{Sgn}(x)$. Since the $\ell_1$ norm is regular, the Riemannian subgradient[3] of (3) concerning only inliers is

$$(\boldsymbol{I} - \boldsymbol{b}\boldsymbol{b}^\top) \sum_{j=1}^{N_k} d_j^k \, \mathrm{sgn}(\boldsymbol{x}_j^{k\top} \boldsymbol{b}) \boldsymbol{x}_j^k \tag{7}$$

Now, we can decompose $\boldsymbol{b}$ by

$$\boldsymbol{b} = \sin(\theta_k)\boldsymbol{s}_k + \cos(\theta_k)\boldsymbol{n}_k, \tag{8}$$

where $\boldsymbol{s}_k := \boldsymbol{\Pi}_{\mathcal{H}_k}(\boldsymbol{b})/\|\boldsymbol{\Pi}_{\mathcal{H}_k}(\boldsymbol{b})\|_2$, $\boldsymbol{n}_k := \boldsymbol{\Pi}_{\mathcal{H}_k^\perp}(\boldsymbol{b})/\|\boldsymbol{\Pi}_{\mathcal{H}_k^\perp}(\boldsymbol{b})\|_2$, and $\theta_k$ is the principal angle from $\boldsymbol{b}$ to $\mathcal{H}_k^\perp = \mathrm{Span}(\boldsymbol{b}_k^*)$. With this, it follows from [72, Lemma 18] that (7) has its $\ell_2$ norm squared

$$\left\| (\boldsymbol{I} - \boldsymbol{b}\boldsymbol{b}^\top) \sum_{j=1}^{N_k} d_j^k \, \mathrm{sgn}(\boldsymbol{x}_j^{k\top} \boldsymbol{b}) \boldsymbol{x}_j^k \right\|_2^2$$

$$= \left\| \sum_{j=1}^{N_k} d_j^k \, \mathrm{sgn}(\boldsymbol{x}_j^{k\top} \boldsymbol{b}) \boldsymbol{x}_j^k \right\|_2^2 - \sin^2(\theta_k) \left( \sum_{j=1}^{N_k} d_j^k |\boldsymbol{x}_j^{k\top} \boldsymbol{s}_k| \right)^2. \tag{9}$$

**Lower bound of (9).** Note that the first term in (9) can be lower bounded by

---

[3]The definition of *regular function* and *Riemannian subgradient* follows from Definition 5.1 and Theorem 5.1 of [86].

$$\left\| \sum_{j=1}^{N_k} d_j^k \, \mathrm{sgn}(\boldsymbol{x}_j^{k\top} \boldsymbol{b}) \boldsymbol{x}_j^k \right\|_2^2 \tag{10}$$

$$= \left\| \sum_{j=1}^{N_k} d_j^k \, \mathrm{sgn}(\boldsymbol{x}_j^{k\top} \boldsymbol{b}) \boldsymbol{x}_j^k \right\|_2^2 \| \boldsymbol{s}_k \|_2^2 \tag{11}$$

$$\geq \left( \sum_{j=1}^{N_k} d_j^k \, \mathrm{sgn}(\boldsymbol{x}_j^{k\top} \boldsymbol{b}) \boldsymbol{x}_j^{k\top} \boldsymbol{s}_k \right)^2 \tag{12}$$

$$= \left( \sum_{j=1}^{N_k} d_j^k \, \mathrm{sgn}(\boldsymbol{x}_j^{k\top} \boldsymbol{s}_k) \boldsymbol{x}_j^{k\top} \boldsymbol{s}_k \right)^2 \tag{13}$$

$$= \left( \sum_{j=1}^{N_k} d_j^k \big| \boldsymbol{x}_j^{k\top} \boldsymbol{s}_k \big| \right)^2 . \tag{14}$$

Here (12) is due to Cauchy-Schwarz. Therefore, we have

$$(9) \geq \cos^2(\theta_k) \left( \sum_{j=1}^{N_k} d_j^k \big| \boldsymbol{x}_j^{k\top} \boldsymbol{s}_k \big| \right)^2 \tag{15}$$

$$\geq \cos^2(\theta_k) N_k^2 c_{\mathrm{in},k,\min}^2, \tag{16}$$

where (16) follows from the definition of $c_{\mathrm{in},k,\min}$ in (2).

**Size of the Riemannian subgradient.** Recall that we assuemd $\boldsymbol{b}$ is a Riemannian critical point of (3) that is not orthogonal to $\mathcal{H}_k$. From Assumption A.1, we have that $\boldsymbol{b}$ is orthogonal to at most $R \leq D - 1$ outliers. Therefore, any Riemannian subgradient of (3) evaluated at $\boldsymbol{b}$ must satisfy

$$\boldsymbol{0} = (\boldsymbol{I} - \boldsymbol{b}\boldsymbol{b}^\top) \left( \sum_{j=1}^{N_k} d_j^k \, \mathrm{sgn}(\boldsymbol{x}_j^{k\top} \boldsymbol{b}) \boldsymbol{x}_j^k + \sum_{j=1}^{M} q_j^k \, \mathrm{sign}(\boldsymbol{o}_j^\top \boldsymbol{b}) \boldsymbol{o}_j + \boldsymbol{\xi} \right), \tag{17}$$

where $\boldsymbol{\xi} = \sum_{j=1}^{R} \beta_j q_j^k \boldsymbol{o}_{l_j}$ such that $\{\boldsymbol{o}_{l_j}\}_{j=1}^{R}$ are $R \leq D - 1$ outliers from $\{\boldsymbol{o}_i\}_{i=1}^{M}$ and $\beta_j \in [-1, 1]$ for $j = 1, \ldots, R$. Taking $\ell_2$ norm on both sides and using the triangular inequality $\|a\| \geq \|a - b\| - \|b\|$, we have

$$0 \geq \left\| (\boldsymbol{I} - \boldsymbol{b}\boldsymbol{b}^\top) \sum_{j=1}^{N_k} d_j^k \, \mathrm{sgn}(\boldsymbol{x}_j^{k\top} \boldsymbol{b}) \boldsymbol{x}_j^k \right\|_2 - \left\| (\boldsymbol{I} - \boldsymbol{b}\boldsymbol{b}^\top) \sum_{j=1}^{M} q_j^k \, \mathrm{sign}(\boldsymbol{o}_j^\top \boldsymbol{b}) \boldsymbol{o}_j \right\|_2 - \left\| (\boldsymbol{I} - \boldsymbol{b}\boldsymbol{b}^\top) \sum_{j=1}^{R} \beta_j q_j^k \boldsymbol{o}_{l_j} \right\|_2 . \tag{18}$$

Combining this with (16), we have

$$\cos(\theta_k) N_k c_{\mathrm{in},k,\min} \tag{19}$$

$$\leq \left\| (\boldsymbol{I} - \boldsymbol{b}\boldsymbol{b}^\top) \sum_{j=1}^{N_k} d_j^k \, \mathrm{sgn}(\boldsymbol{x}_j^{k\top} \boldsymbol{b}) \boldsymbol{x}_j^k \right\|_2 \tag{20}$$

$$\leq \left\| (\boldsymbol{I} - \boldsymbol{b}\boldsymbol{b}^\top) \sum_{j=1}^{M} q_j^k \, \mathrm{sign}(\boldsymbol{o}_j^\top \boldsymbol{b}) \boldsymbol{o}_j \right\|_2 + \left\| (\boldsymbol{I} - \boldsymbol{b}\boldsymbol{b}^\top) \sum_{j=1}^{R} \beta_j q_j^k \boldsymbol{o}_{l_j} \right\|_2 \tag{21}$$

$$\leq M \eta_{\mathrm{out},k} + D = M \bar{\eta}_{\mathrm{out},k}, \tag{22}$$

where the first inequality is due to (16), second due to (18). This is equivalent to saying

$$\theta_k \geq \arcsin \sqrt{1 - \left( \frac{M \bar{\eta}_{\mathrm{out},k}}{N_k c_{\mathrm{in},k,\min}} \right)^2} . \tag{23}$$

That is, a Riemannian critical point $b$ of (3) is either $\pm b_k^*$, or $b$ must have a principal angle from $\mathcal{H}_k$ smaller than or equal to $\arcsin\left(\frac{M\bar{\eta}_{\text{out},k}}{N_k c_{\text{in},k,\min}}\right)$. This completes the proof. $\qquad\square$

With Theorem 4.2 characterizing critical points of the coordinate-wise problem (3), we are now ready to prove Theorem 4.3 that studies the minimizers of (3). For convenience, we define

$$c_{\text{out},k,\min} := \frac{1}{M} \min_{b \in \mathbb{S}^{D-1}} \sum_{j=1}^{M} q_j^k |o_j^\top b| \tag{24}$$

$$c_{\text{out},k,\max} := \frac{1}{M} \max_{b \in \mathbb{S}^{D-1}} \sum_{j=1}^{M} q_j^k |o_j^\top b| \tag{25}$$

$$\eta_{\text{out},k} := \frac{1}{M} \max_{b \in \mathbb{S}^{D-1}} \left\| (I - bb^\top) \sum_{j=1}^{M} q_j^k \operatorname{sgn}(o_j^\top b) o_j \right\|_2, \tag{26}$$

where one notes that $c_{\text{out},k} = c_{\text{out},k,\max} - c_{\text{out},k,\min}$ and $\bar{\eta}_{\text{out},k} := \eta_{\text{out},k} + \frac{D}{M}$.

**Theorem 4.3.** *Any minimizer of* (3) *must be* $\pm b_k^*$ *if*

$$\frac{M}{N_k} \frac{\sqrt{c_{\text{out},k}^2 + \bar{\eta}_{\text{out},k}^2}}{c_{\text{in},k,\min}} < 1. \tag{4}$$

*Further, if* (4) *holds for* $k = 1, \ldots, K$, *then* $\{\pm b_k^*\}_{k=1}^{K}$ *are coordinate-wise minimizers of* (GPCA-$\ell_1$).

*Proof.* Let $b$ now be a minimizer of (3). If $b = \pm b_k^*$ then we are done. Suppose the contrary, i.e., $b \neq \pm b_k^*$, and let $\theta_k$ denotes the principal angle from $b$ to $\mathcal{H}_k^\perp = \operatorname{Span}(b_k^*)$. Since $b$ is a solution, it must be a Riemannian critical point, therefore by Theorem 4.2 we must have

$$\cos(\theta_k) \leq \frac{M\bar{\eta}_{\text{out},k}}{N_k c_{\text{in},k,\min}}, \tag{27}$$

i.e., $b$ is far away from $\mathcal{H}_k^\perp$. We will now show that in fact $b$ can not be far away from $\mathcal{H}_k^\perp$. Again taking a decomposition of $b$ as in (8), we have

$$g_k(b) = \sum_{j=1}^{N_k} d_j^k |x_j^{k\top} b| + \sum_{j=1}^{M} q_j^k |o_j^\top b| \tag{28}$$

$$= \sin(\theta_k) \sum_{j=1}^{N_k} d_j^k |x_j^{k\top} s| + \sum_{j=1}^{M} q_j^k |o_j^\top b| \tag{29}$$

$$\geq \sin(\theta_k) N_k c_{\text{in},k,\min} + M c_{\text{out},k,\min}. \tag{30}$$

On the other hand, since $b$ is a minimizer, it holds that

$$g_k(b) = \min_{b \in \mathbb{S}^{D-1}} g_k(b) \tag{31}$$

$$\leq \min_{b \in \mathbb{S}^{D-1}, b \perp \mathcal{H}_k} g_k(b) \tag{32}$$

$$= \min_{b \in \mathbb{S}^{D-1}, b \perp \mathcal{H}_k} \sum_{j=1}^{M} q_j^k |o_j^\top b| \tag{33}$$

$$\leq M c_{\text{out},k,\max}. \tag{34}$$

Combining (30) and (34) yields

$$\sin(\theta_k) \leq \frac{M}{N_k} \frac{c_{\text{out},k,\max} - c_{\text{out},k,\min}}{c_{\text{in},k,\min}}. \tag{35}$$

To show that this leads to a contradiction, we observe from (27) and (35) that

$$1 = \cos^2(\theta_k) + \sin^2(\theta_k) \tag{36}$$

$$\leq \left(\frac{M}{N_k}\right)^2 \frac{(c_{\text{out},k,\max} - c_{\text{out},k,\min})^2 + \bar{\eta}_{\text{out},k}^2}{c_{\text{in},k,\min}^2} \tag{37}$$

$$< 1, \tag{38}$$

where the last line follows from the assumption in the theorem. This completes the proof. □

## A.2. Convergence Analysis

**Theorem 5.2.** *Under Assumption 5.1 (A1), any limit point of the sequence $\{(\boldsymbol{b}_1^{(t)}, \ldots, \boldsymbol{b}_K^{(t)})\}_t$ generated by the HARD-$\ell_1$ algorithm is a coordinate-wise minimizer of (GPCA-$\ell_1$).*

*Proof.* Define the objective function of (GPCA-$\ell_1$) to be $f(\boldsymbol{b}_1, \ldots, \boldsymbol{b}_K) := \sum_{j=1}^{N+M} \prod_{k=1}^{K} |\tilde{\boldsymbol{x}}_j^\top \boldsymbol{b}_k|$. By the definition of HARD-$\ell_1$ (Algorithm 1), we have

$$f(\boldsymbol{b}_1^{(t+1)}, \ldots, \boldsymbol{b}_K^{(t+1)}) \leq f(\boldsymbol{b}_1^{(t+1)}, \ldots, \boldsymbol{b}_K^{(t)})$$
$$\leq \cdots$$
$$\leq f(\boldsymbol{b}_1^{(t)}, \ldots, \boldsymbol{b}_K^{(t)})$$

Since $f$ is lower bounded (by 0), $f(\boldsymbol{b}_1^{(t)}, \ldots, \boldsymbol{b}_K^{(t)})$ must converge to some non-negative number $\overline{f}$. On the other hand, since $\{(\boldsymbol{b}_1^{(t)}, \ldots, \boldsymbol{b}_K^{(t)})\}_t$ is a bounded sequence, it must have a convergent subsequence, say $\{(\boldsymbol{b}_1^{(t_i)}, \ldots, \boldsymbol{b}_K^{(t_i)})\}_i$. Let $(\overline{\boldsymbol{b}}_1, \ldots, \overline{\boldsymbol{b}}_K)$ be the limit of $\{(\boldsymbol{b}_1^{(t_i)}, \ldots, \boldsymbol{b}_K^{(t_i)})\}_i$. Furthermore, passing to subsequences if needed, we can assume that $\{(\boldsymbol{b}_1^{(t_i+1)}, \ldots, \boldsymbol{b}_k^{(t_i+1)}, \boldsymbol{b}_{k+1}^{(t_i)}, \ldots, \boldsymbol{b}_K^{(t_i)})\}_i$ converges to $(\overline{\boldsymbol{a}}_1, \ldots, \overline{\boldsymbol{a}}_k, \overline{\boldsymbol{b}}_{k+1}, \ldots, \overline{\boldsymbol{b}}_K)$. Since the sphere is a closed set, all $\overline{\boldsymbol{a}}_k$ and $\overline{\boldsymbol{b}}_k$ are unit vectors on the sphere.

We now prove $\overline{\boldsymbol{a}}_j = \overline{\boldsymbol{b}}_j$ for every $j$. First of all, for $j = 1$ and for every unit vector $\boldsymbol{b} \in \mathbb{S}^{D-1}$, we have

$$f(\boldsymbol{b}_1^{(t_i+1)}, \boldsymbol{b}_2^{(t_i)} \ldots, \boldsymbol{b}_K^{(t_i)}) \leq f(\boldsymbol{b}, \boldsymbol{b}_2^{(t_i)}, \ldots, \boldsymbol{b}_K^{(t_i)}).$$

Since $f$ is continuous, setting $t_i \to \infty$ yields

$$\overline{f} = f(\overline{\boldsymbol{a}}_1, \overline{\boldsymbol{b}}_2, \ldots, \overline{\boldsymbol{b}}_K) \leq f(\boldsymbol{b}, \overline{\boldsymbol{b}}_2, \ldots, \overline{\boldsymbol{b}}_K)$$

for every $\boldsymbol{b} \in \mathbb{S}^{D-1}$. Since $f(\overline{\boldsymbol{b}}_1, \overline{\boldsymbol{b}}_2, \ldots, \overline{\boldsymbol{b}}_K)$ is also equal to $\overline{f}$ and (A1) has a unique minimizer, we get $\overline{\boldsymbol{a}}_1 = \overline{\boldsymbol{b}}_1$. Then, a similar argument that passes $t_i$ to limits would yield

$$\overline{f} = f(\overline{\boldsymbol{b}}_1, \overline{\boldsymbol{a}}_2, \overline{\boldsymbol{b}}_3, \ldots, \overline{\boldsymbol{b}}_K)$$
$$= f(\overline{\boldsymbol{a}}_1, \overline{\boldsymbol{a}}_2, \overline{\boldsymbol{b}}_3, \ldots, \overline{\boldsymbol{b}}_K)$$
$$\leq f(\overline{\boldsymbol{a}}_1, \boldsymbol{b}, \overline{\boldsymbol{b}}_3, \ldots, \overline{\boldsymbol{b}}_K)$$
$$= f(\overline{\boldsymbol{b}}_1, \boldsymbol{b}, \overline{\boldsymbol{b}}_3, \ldots, \overline{\boldsymbol{b}}_K)$$

for every $\boldsymbol{b} \in \mathbb{S}^{D-1}$. And a similar argument with (A1) proves $\overline{\boldsymbol{b}}_2 = \overline{\boldsymbol{a}}_2$. Generalizing, we can prove $\overline{\boldsymbol{a}}_j = \overline{\boldsymbol{b}}_j$ for every $j$, which implies $(\overline{\boldsymbol{b}}_1, \ldots, \overline{\boldsymbol{b}}_K)$ is indeed a coordinate-wise minimizer of (GPCA-$\ell_1$). □

**Theorem 5.3.** *Under Assumption 5.1 (A2), any limit point of the iterates $\{(\boldsymbol{b}_1^{(t)}, \ldots, \boldsymbol{b}_K^{(t)})\}_t$ generated by the HARD-Huber algorithm is a coordinate-wise minimizer and also a critical point of (GPCA-Huber).*

*Proof.* One can prove convergence to coordinate-wise minimizers using the same idea as we have shown in the proof of Theorem 5.2, therefore we omit the details here. Let $(\overline{\boldsymbol{b}}_1, \ldots, \overline{\boldsymbol{b}}_K)$ be such limit point (and coordinate-wise minimizer), and we will show it is a critical point of (GPCA-Huber). Note that each $\overline{\boldsymbol{b}}_k$ satisfies the optimality condition that the Riemannian gradient of $H_k(\boldsymbol{b}) := \sum_{j=1}^{N+M} h(\tilde{\boldsymbol{x}}_j^\top \boldsymbol{b}) \prod_{i \neq k}^{K} h(\tilde{\boldsymbol{x}}_j^\top \overline{\boldsymbol{b}}_i)$ at $\overline{\boldsymbol{b}}_k$ is zero, that is,

$$\left(\boldsymbol{I} - \overline{\boldsymbol{b}}_k \overline{\boldsymbol{b}}_k^\top\right) \nabla H_k(\overline{\boldsymbol{b}}_k) = 0.$$

Here $\boldsymbol{I}$ is the identity matrix of size $D \times D$. Summing up these equalities over $k$, we arrive at the fact that the Riemannian gradient of the objective of (GPCA-Huber) at $(\overline{\boldsymbol{b}}_1, \ldots, \overline{\boldsymbol{b}}_K)$ is equal to zero, meaning that $(\overline{\boldsymbol{b}}_1, \ldots, \overline{\boldsymbol{b}}_K)$ is a critical point of (GPCA-Huber). □

**Theorem 5.4.** *Under Assumption 5.1 (A3), any limit point of the iterates $\{(\boldsymbol{b}_1^{(t)}, \ldots, \boldsymbol{b}_K^{(t)})\}_t$ generated by the HARD-Huber+ algorithm is a critical point of (GPCA-Huber).*

*Proof.* Write $\boldsymbol{b}_{1:K}$ for the collection of vectors $\{\boldsymbol{b}_k\}_{k=1}^K$. Define $H(\boldsymbol{b}_{1:K}) := \sum_{j=1}^{N+M} \prod_{k=1}^K h(\tilde{\boldsymbol{x}}_j^\top \boldsymbol{b}_k)$. Define a surrogate function $u$ of $h(\cdot)$ via

$$u(r; s) := h(s) + \frac{r^2 - s^2}{2 \cdot \max\{|s|, \delta\}}$$

and surrogate functions ($\forall i = 1, \ldots, K$)

$$U_k(\boldsymbol{z}; \boldsymbol{b}_{1:K}) := \sum_{j=1}^{N+M} u(\tilde{\boldsymbol{x}}_j^\top \boldsymbol{z}; \tilde{\boldsymbol{x}}_j^\top \boldsymbol{b}_k) \cdot \prod_{i \neq k} h(\tilde{\boldsymbol{x}}_j^\top \boldsymbol{b}_i). \tag{39}$$

One verifies $u(r, s) \geq h(r)$ for any $r$ and $s$, which means $U_k(\boldsymbol{z}; \boldsymbol{b}_{1:K}) \geq H(\boldsymbol{b}_{1:k-1}, \boldsymbol{z}, \boldsymbol{b}_{k+1:K})$ for any $k, \boldsymbol{z}$, and $\boldsymbol{b}_{1:K}$, with the equality attained when $\boldsymbol{z} = \boldsymbol{b}_k$.

Define $\boldsymbol{q}_1^{(t)} := \boldsymbol{b}_{1:K}^{(t)}$, and for $k = 2, \ldots, K$, define

$$\boldsymbol{q}_k^{(t)} := \left(\boldsymbol{b}_{1:k-1}^{(t+1)}, \boldsymbol{b}_{k:K}^{(t)}\right) \tag{40}$$

By these definitions, we can write (Huber+) as

$$\begin{aligned}
\boldsymbol{b}_k^{(t+1)} &= \underset{\boldsymbol{b}_k \in \mathbb{S}^{D-1}}{\operatorname{argmin}} \sum_{j=1}^{N+M} v_{j,k}^{(t)} \cdot \frac{(\tilde{\boldsymbol{x}}_j^\top \boldsymbol{b}_k)^2}{\max\{|\tilde{\boldsymbol{x}}_j^\top \boldsymbol{b}_k^{(t)}|, \delta\}} \\
&= \underset{\boldsymbol{b}_k \in \mathbb{S}^{D-1}}{\operatorname{argmin}} U_k(\boldsymbol{b}_k; \boldsymbol{q}_k^{(t)}),
\end{aligned} \tag{41}$$

which implies for every $t$ and $k$ that

$$U_k(\boldsymbol{b}_k^{(t+1)}; \boldsymbol{q}_k^{(t)}) \leq U_k(\boldsymbol{b}_k; \boldsymbol{q}_k^{(t)}), \quad \forall \boldsymbol{b}_k \in \mathbb{S}^{D-1}. \tag{42}$$

Since $\{\boldsymbol{b}_{1:K}^{(t)}\}_t$ is bounded, it must contain a subsequence $\{\boldsymbol{b}_{1:K}^{(t_i)}\}_i$ convergent to some limit point, which we denote by $\overline{\boldsymbol{b}}_{1:K}$. Each $\overline{\boldsymbol{b}}_k$ lies in $\mathbb{S}^{D-1}$ as $\mathbb{S}^{D-1}$ is closed. We need to show $\overline{\boldsymbol{b}}_{1:K}$ is a critical point of (GPCA-Huber). Passing to subsequences if necessary, let us assume $\{\boldsymbol{q}_k^{(t_i)}\}_{t_i} = \{(\boldsymbol{b}_1^{(t_i+1)}, \ldots, \boldsymbol{b}_{k-1}^{(t_i+1)}, \boldsymbol{b}_k^{(t_i)}, \ldots, \boldsymbol{b}_K^{(t_i)})\}_i$ converges to $(\overline{\boldsymbol{a}}_1, \ldots, \overline{\boldsymbol{a}}_{k-1}, \overline{\boldsymbol{b}}_k, \ldots, \overline{\boldsymbol{b}}_K) =: \overline{\boldsymbol{q}}_k$.

From the above discussions, it follows that

$$\begin{aligned}
H(\boldsymbol{b}_{1:K}^{(t_i+1)}) &\leq U_K(\boldsymbol{b}_K^{(t_i+1)}; \boldsymbol{q}_K^{(t_i)}) \\
&\overset{(41)}{\leq} U_K(\boldsymbol{b}_K^{(t_i)}; \boldsymbol{q}_K^{(t_i)}) \\
&= H(\boldsymbol{q}_K^{(t_i)}) \\
&\leq U_{K-1}(\boldsymbol{b}_{K-1}^{(t_i+1)}; \boldsymbol{q}_{K-1}^{(t_i)}) \\
&\leq \cdots \\
&\leq H(\boldsymbol{b}_{1:K}^{(t_i)})
\end{aligned} \tag{43}$$

This implies the sequence $\{H(\boldsymbol{b}_{1:K}^{(t_i)})\}_i$ is non-increasing, thus convergent, say to $\overline{H}$. Since $H$ and $U_k$'s are continuous, letting $t_i \to \infty$, (43) implies

$$U_k(\overline{\boldsymbol{a}}_k; \overline{\boldsymbol{q}}_k) = U_K(\overline{\boldsymbol{b}}_k; \overline{\boldsymbol{q}}_k) = \overline{H}, \forall k = 1, \ldots, K. \tag{44}$$

But substituting $t = t_i$ and then $t_i \to \infty$ into (42) shows that $\overline{\boldsymbol{a}}_k$ is a global minimizer of $U_k(\boldsymbol{b}_k; \overline{\boldsymbol{q}}_k)$ in variable $\boldsymbol{b}_k \in \mathbb{S}^{D-1}$. It then follows from (44) that $\overline{\boldsymbol{a}}_k$ and $\overline{\boldsymbol{b}}_k$ are both minimizers, and from the unique minimizer assumption that $\overline{\boldsymbol{a}}_k = \overline{\boldsymbol{b}}_k$ for every $k = 1, \ldots, K$. Thus $\overline{\boldsymbol{q}}_k = \overline{\boldsymbol{b}}_{1:K}$.

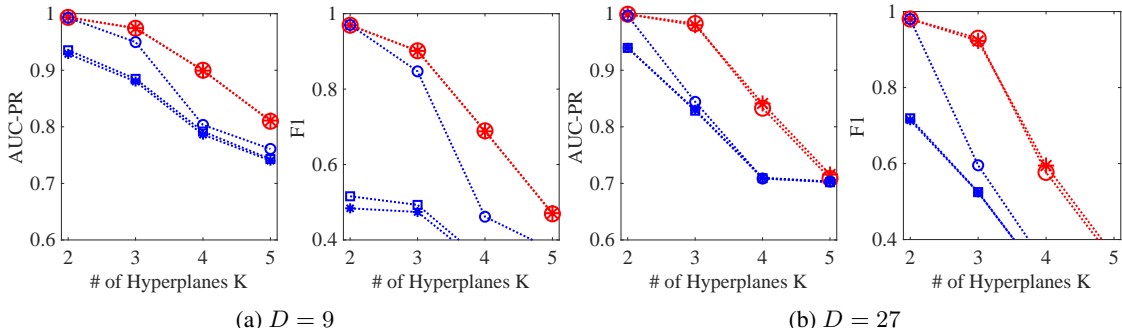

(a) $D = 9$        (b) $D = 27$

Figure 4: Area under the precision-recall curve and F1 score of different methods on synthetic data (§6.1) over 100 repeated experiments. Here we fix an outlier ratio of $M/(N + M) = 30\%$, vary $K$ in the $x$-axis, and $D$ in different sub-figures.

Thus $\overline{\boldsymbol{b}}_k$ is a critical point of the optimization problem of minimizing $U_k(\boldsymbol{b}_k; \overline{\boldsymbol{b}}_{1:K})$ in variable $\boldsymbol{b}_k \in \mathbb{S}^{D-1}$. The critical point is in the Riemannian sense, i.e.,

$$\left(\boldsymbol{I} - \overline{\boldsymbol{b}}_k \overline{\boldsymbol{b}}_k^\top\right) \nabla_{\overline{\boldsymbol{b}}_k} U(\overline{\boldsymbol{b}}_k; \overline{\boldsymbol{b}}_{1:K}) = 0, \ \forall k = 1, \dots K,$$

where $\boldsymbol{I}$ is the identity matrix. Since $\nabla_{\overline{\boldsymbol{b}}_k}(\overline{\boldsymbol{b}}_k; \overline{\boldsymbol{b}}_{1:K}) = \nabla_{\overline{\boldsymbol{b}}_k} H(\overline{\boldsymbol{b}}_{1:K})$, the above condition becomes

$$\left(\boldsymbol{I} - \overline{\boldsymbol{b}}_k \overline{\boldsymbol{b}}_k^\top\right) \nabla_{\overline{\boldsymbol{b}}_k} H(\overline{\boldsymbol{b}}_{1:K}) = 0, \ \forall k = 1, \dots K.$$

Summing the above equations over $k$ shows that the limit point $\overline{\boldsymbol{b}}_{1:K}$ satisfies the first-order Riemannian optimality condition of (GPCA-Huber), so the proof is complete. $\qquad \square$

## B. Additional Experiments

### B.1. Outlier Rejection Performance

**Fixed $M/(N + M)$, varying $K, D$.** While the synthetic experiments in §6.1 compares the clustering performance, we further shed light on the outlier rejection aspect. Figure 4 reports AUC-ROC and F1-score over 100 repeated experiments. Notably when $D = 27$ and $K = 3$, HARD surpasses KH-DPCP by a large margin: HARD-{$\ell_1$+, Huber+} yield at least 0.97 AUC-PR, as opposed to 0.84 for KH-DPCP. Based on our investigations, KH-DPCP converges[4] to a point whose objective value is 106% higher than that of the true hyperplane arrangement. In contrast, HARD-$\ell_1$+ and HARD-Huber+ are able to achieve an objective value within 9% and 13% of (GPCA-$\ell_1$) and (GPCA-Huber) evaluated at the true arrangement, respectively. In retrospect, our algorithms outperform KH-DPCP for two possible reasons. First, the objective of KH-DPCP is highly non-convex while (GPCA-$\ell_1$) is marginally convex and (GPCA-Huber) marginally convex and smooth; in this sense, our formulations are more amenable to optimization. Second, KH-DPCP assigns points to the hyperplanes in a 'hard' manner, while HARD-$\ell_1$+ and HARD-Huber use 'soft' weights as in ($\ell_1$+) and (Huber+); soft weights allow for uncertain assignments, which might be less sensitive to poor initialization.

**Fixed $K$, varying $D, M/(N + M)$.** Now we stick to $K = 3$ and vary the outlier ratio $M/(N + M) \in \{0 : 0.1 : 0.5\}$ and $D \in \{9, 27\}$. Figure 5 shows clustering accuracy and AUC-PR over 100 repeated experiments. As a first observation, as the outlier ratio goes higher all methods degrade in performance. Again, HARD is consistently the most accurate and robust in most cases. For example, HARD methods yield at least 0.97 AUC-PR with $\leq 30\%$ outlier ratio.

---

[4]We used either projected Riemannian subgradient descent [7] or iteratively reweighted least squares [6] to solve the (DPCP) problem inside the KH framework, and the conclusion is the same.

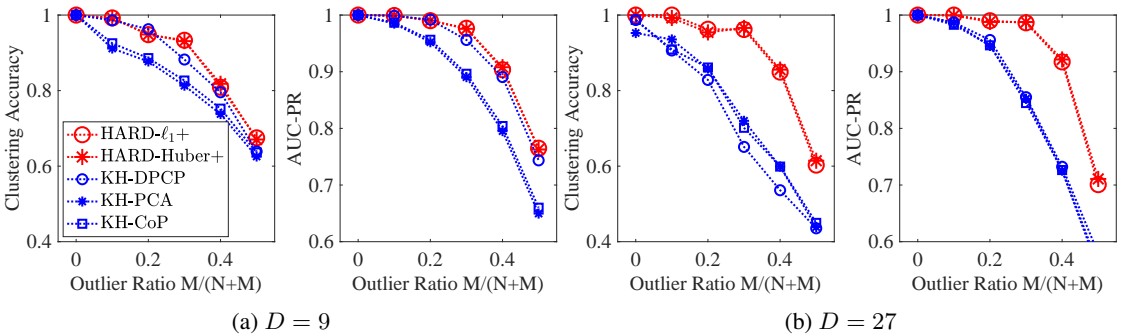

Figure 5: Clustering accuracy and area under the precision-recall curve of different methods on synthetic data (§6.1) over 100 repeated experiments. We fix $K = 3$, vary outlier ratio $M/(N + M)$ in the $x$-axis, and $D$ in different sub-figures.

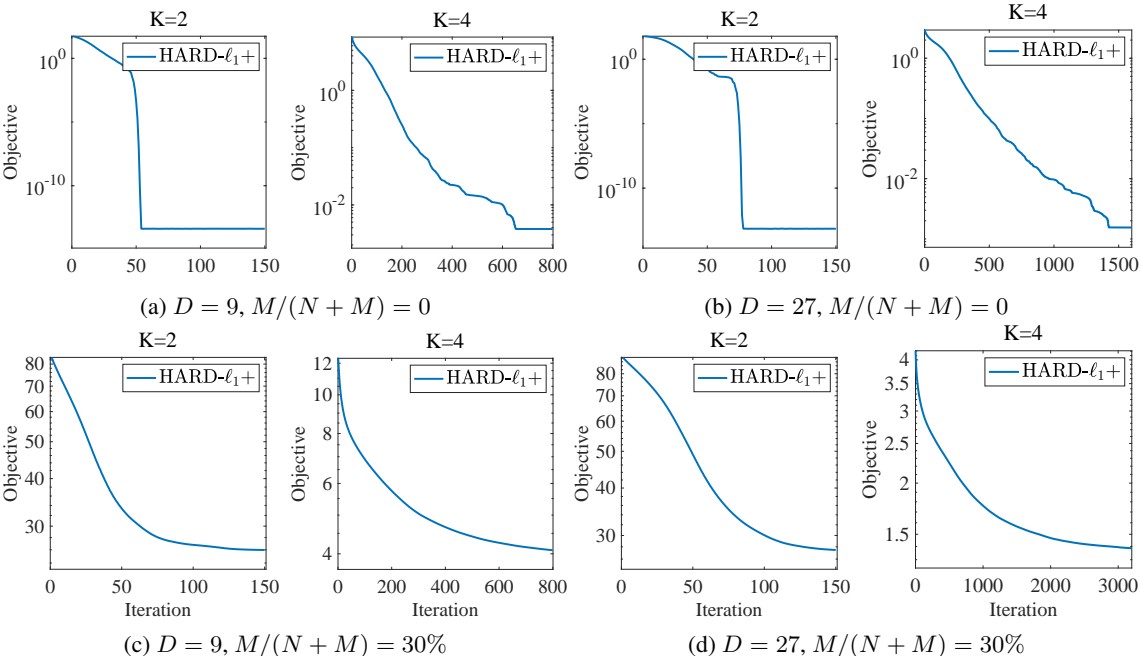

Figure 6: Objective value of (GPCA-$\ell_1$) versus HARD-$\ell_1$+ iterations over 100 repeated experiments.

## B.2. Empirical Evidence of HARD-$\ell_1$+ Does Not Increase (GPCA-$\ell_1$) Objective

While the proof of Theorem 5.4 shows that HARD-Huber+ does not increase the (GPCA-Huber) objective, we have not proved its counterpart for HARD-$\ell_1$+ and (GPCA-$\ell_1$). Nevertheless, here we give empirical evidence on why the latter is expected to be true. To do so, we run HARD-$\ell_1$+ using the setup in §6.1 with $D \in \{9, 27\}$, $K \in \{2, 4\}$ and $M/(N + M) \in \{0, 0.3\}$. We plot in Figure 6 the objective value of (GPCA-$\ell_1$) each time ($\ell_1$+) is executed. For comparison, we do the same for (GPCA-Huber) and (Huber+) in Figure 7. $\delta$ is set to $10^{-16}$. As seen, in all settings HARD-$\ell_1$+ never increases the (GPCA-$\ell_1$) objective.

## B.3. Effect of Multiple Initializations

Since KH and HARD are sensitive to initialization, we also study the effect of the number of random initializations. Specifically, we generate one random hyperplane arrangement and use that as initialization for all methods. This is repeated multiple times, and for each method the estimated arrangement with the smallest

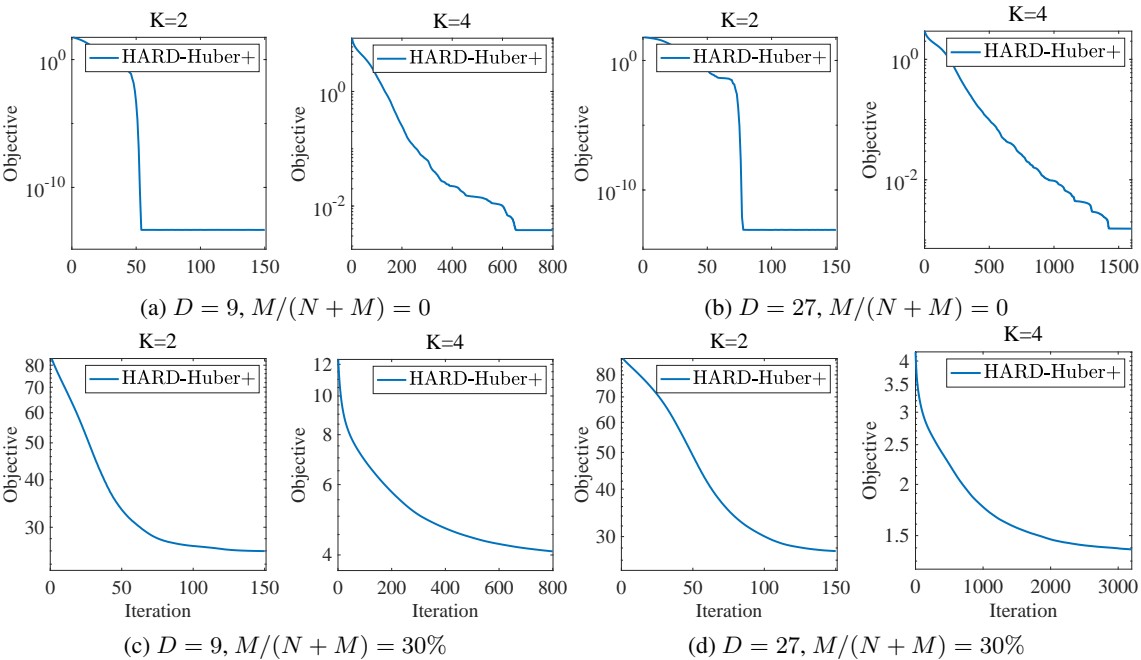

Figure 7: Objective value (GPCA-Huber) versus HARD-Huber+ iterations over 100 repeated experiments.

objective value[5] is kept as the final output. Figure 8 reports clustering accuracy and the final objective relative to ground-truth using $\{1, 5, 9\}$ initializations with $D = 27$ and $K \in \{2, 3, 4, 5\}$. To begin with, all methods yield higher accuracy and lower final objective as expected. Notably, when $K = 3$ HARD methods yield 100% accuracy with only 5 initializations, while KH-DPCP gives less than 95% accuracy with 9 initializations. Compared to HARD, KH methods more easily converge at an objective value higher than that of the true hyperplane arrangement, an observation consistent with §6.1.

## B.4. CIFAR-10 without Projection

To further see how the methods perform on real data when $D$ is large, we conduct the following experiment. We take the features on CIFAR-10 (§6.2), which are $5 \cdot 10^4$ points in $\mathbb{S}^{128}$. We consider them directly and do not pre-process them using projection, in contrast to the experiment in §6.2. Since points from each ground-truth cluster lie close to a low-dimensional subspace, all the points are approximately contained in two hyperplanes.[6] Therefore, we perform hyperplane clustering on these points to extract *two* hyperplanes. We report the *separation rate*, which is simply clustering accuracy (§6) normalized to have a maximum value of 100%. A separation rate of 100% means that there are two ground-truth clusters, each contained in an estimated cluster. Figure 9 reports separation rate and running time over 30 runs of the methods. As seen, HARD-$\{\ell_1+,$Huber+$\}$ achieve the highest separation rate, are more stable, and use the least time.

## C. Connection and Extension

### C.1. Connection with KH-DPCP

KH-DPCP is similar to the classic $K$-means. It estimates $K$ hyperplanes and the membership of points to them, by alternating between **Hyperplane Estimation** and **Membership Estimation** (detailed below).

---

[5]Since KH-CoP does not have an objective related to the method and that of KH-PCA is not robust to outliers, the objective of KH-DPCP is used for the purpose of selecting a final arrangement. This is also done in [7].

[6]Suppose $\bigoplus_{i=1}^{L} \mathcal{S}_i$ is a direct sum of independent and non-trivial subspaces in $\mathbb{R}^D$. Then $\bigoplus_{i \neq 1} \mathcal{S}_i$ has dimension at most $D - 1$, so it is contained in a hyperplane; similarly for $\bigoplus_{i \neq 2} \mathcal{S}_i$. This is roughly the case of the features used in this experiment, since they are expected to lie in a union of independent subspaces.

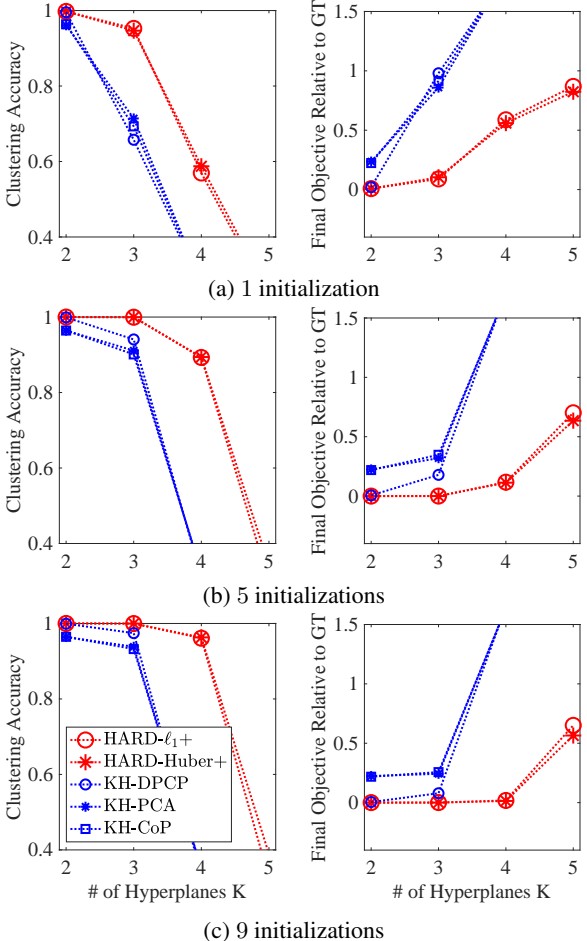

Figure 8: Clustering accuracy and relative optimality of robust hyperplane clustering methods on synthetic data (§6.1) over 100 repeated experiments. Here we fix $D = 27$, vary $K$ in the $x$-axis, and the number of initializations for each method in different rows.

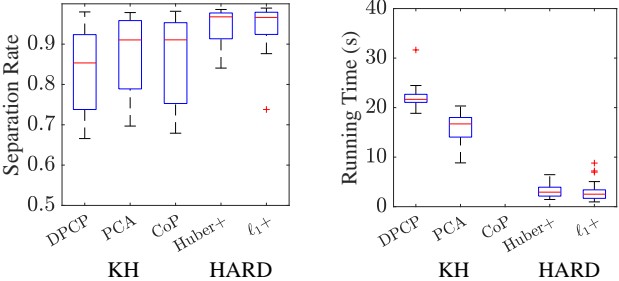

Figure 9: Separation rate and running time of different methods on deep features in $\mathbb{R}^{128}$ extracted from $5 \cdot 10^4$ images from CIFAR-10. For each metric, statistics are shown over 30 runs of the methods. Notably, the proposed HARD-$\{\ell_1+,$Huber+$\}$ achieve the highest separation rate, are more stable and use the least time.

1. **Hyperplane Estimation**: Given the membership of points to hyperplanes, estimate the hyperplanes

$$\forall k = 1, \dots, K, \quad \boldsymbol{b}_k \in \operatorname*{argmin}_{\boldsymbol{b}_k \in \mathbb{S}^{D-1}} \sum_{j=1}^{N+M} w_{jk} \left| \tilde{\boldsymbol{x}}_j^\top \boldsymbol{b}_k \right|, \tag{45}$$

which is the (DPCP) problem [37, 38], solvable by many methods [38, 62–65].

2. **Membership Estimation**: Given the hyperplanes, assign each point to the closest hyperplane

$$\forall k = 1, \dots, K, \quad \forall j = 1, \dots, N+M, \quad w_{jk} = \begin{cases} 1 & \text{if } k = \operatorname*{arg\,min}_{k=1,\dots,K} \left| \tilde{\boldsymbol{x}}_j^\top \boldsymbol{b}_k \right| \\ 0 & \text{otherwise} \end{cases}. \tag{46}$$

The above can be viewed as an alternating minimization algorithm applied to the following objective:

$$\min_{\{\boldsymbol{b}_k\}_{k=1}^K} \sum_{j=1}^{N+M} \min_k \left| \boldsymbol{b}_k^\top \tilde{\boldsymbol{x}}_j \right| \quad \text{s.t.} \quad \boldsymbol{b}_k \in \mathbb{S}^{D-1}, \quad k = 1, \dots, K \tag{47}$$

$$\Leftrightarrow \min_{\{\boldsymbol{b}_k\}_{k=1}^K, \boldsymbol{W}} \sum_{k=1}^K \sum_{j=1}^{N+M} w_{jk} \left| \boldsymbol{b}_k^\top \tilde{\boldsymbol{x}}_j \right| \quad \text{s.t.} \quad \boldsymbol{b}_k \in \mathbb{S}^{D-1}, \quad k = 1, \dots, K, \tag{48}$$

$$\boldsymbol{W} \in \{0,1\}^{(N+M) \times K}, \quad \boldsymbol{W}\boldsymbol{1} = \boldsymbol{1}.$$

Here, $w_{jk}$ is constrained to be a binary number and it is the $(j,k)$-th entry of the matrix $\boldsymbol{W}$.

Beyond the empirical comparison of KH-DPCP and (GPCA-$\ell_1$) in previous experiments (§6 and §B), we note that the two are conceptually related but different. For each point $\tilde{\boldsymbol{x}}_j$, its distances to the $K$ variable hyperplanes are computed, and some function is applied to the $K$ distances to produce one number: the minimum for KH-DPCP and the product for (GPCA-$\ell_1$). This number is zero if and only if $\tilde{\boldsymbol{x}}_j$ belongs to one (or more) of the K hyperplanes.

## C.2. Extension to Clustering Subspaces of Codimension $> 1$

The current paper focuses on hyperplane clustering, which is motivated by many applications on its own (as in §1). Meanwhile, one can extend this idea to clustering data that lie on a union of subspaces $\bigcup_{k=1}^K \mathcal{S}_k \subset \mathbb{R}^D$ of codimension $c > 1$.

**Formulation.** We can directly learn basis *matrices* $\{\boldsymbol{B}_k\}_{k=1}^K$ for the orthogonal complement of the subspaces, by solving the following problem:

$$\min_{\{\boldsymbol{B}_k\}_{k=1}^K} \sum_{j=1}^{N+M} \prod_{k=1}^K \left\| \boldsymbol{B}_k^\top \tilde{\boldsymbol{x}}_j \right\|_2 \quad \text{s.t.} \quad \boldsymbol{B}_k \in \mathbb{O}(D,c), \quad k = 1, \dots, K. \tag{49}$$

Here $\boldsymbol{B}_k$ is constrained to be in $\mathbb{O}(D,c) := \{\boldsymbol{B}_k \in \mathbb{R}^{D \times c} : \boldsymbol{B}_k^\top \boldsymbol{B}_k = \boldsymbol{I}_c\}$, such that the problem is properly normalized and $\left\| \boldsymbol{B}_k^\top \tilde{\boldsymbol{x}}_j \right\|_2$ is the distance of $\tilde{\boldsymbol{x}}_j$ to the $k$-th subspace. An ($\ell_1+$)-like algorithm is

$$w_{j,k}^{(t)} \leftarrow \begin{cases} \prod_{i \neq k} \left\| \tilde{\boldsymbol{x}}_j^\top \boldsymbol{B}_i^{(t)} \right\|_2 & k = 1 \\ \prod_{i<k} \left\| \tilde{\boldsymbol{x}}_j^\top \boldsymbol{B}_i^{(t+1)} \right\|_2 \cdot \prod_{i>k} \left\| \tilde{\boldsymbol{x}}_j^\top \boldsymbol{B}_i^{(t)} \right\|_2 & k > 1 \end{cases}$$

$$\boldsymbol{b}_k^{(t+1)} \in \operatorname*{argmin}_{\boldsymbol{B}_k \in \mathbb{O}(D,c)} \sum_{j=1}^{N+M} w_{j,k}^{(t)} \cdot \frac{(\tilde{\boldsymbol{x}}_j^\top \boldsymbol{B}_k)^2}{\max\left\{ \left\| \tilde{\boldsymbol{x}}_j^\top \boldsymbol{B}_k^{(t)} \right\|_2, \delta \right\}} \tag{50}$$

where the update (50) can again be solved via SVD. Other algorithms can be extended in a similar manner.

**Theoretical Analysis.** For understanding the minimizers, we can extend the quantities in (2) by (re-)defining

$$c_{\text{in},k,\min} := \frac{1}{N_k} \min_{\boldsymbol{b} \in \mathcal{S}_k \cap \mathbb{S}^{D-1}} \sum_{j=1}^{N_k} d_j^k \cdot \left| \boldsymbol{x}_j^{k\top} \boldsymbol{b} \right|$$

$$c_{\text{out},k} := \frac{1}{M} \left( \max_{\boldsymbol{B} \in \mathbb{O}(D,c)} \sum_{j=1}^{M} q_j^k \cdot \left| \boldsymbol{o}_j^\top \boldsymbol{B} \right| - \min_{\boldsymbol{B} \in \mathbb{O}(D,c)} \sum_{j=1}^{M} q_j^k \cdot \left| \boldsymbol{o}_j^\top \boldsymbol{B} \right| \right)$$

$$\bar{\eta}_{\text{out},k} := \frac{1}{M} \max_{\boldsymbol{B} \in \mathbb{O}(D,c)} \left\| (\boldsymbol{I} - \boldsymbol{B}\boldsymbol{B}^\top) \sum_{j=1}^{M} q_j^k \boldsymbol{o}_j \, \text{sign}(\boldsymbol{o}_j^\top \boldsymbol{B}) \right\|_F + \frac{D}{M},$$

and it should not be difficult to extend the results in §4.2. Most proof steps should be similar, with an additional inequality [72, Sublemma 2] to handle $c_{\text{in},k,\min}$ and more care on the principal angles between two subspaces.

Results of similar flavor to those of §5 follow directly. Specifically, Theorem 5.2 can be extended directly without any change of the proof logic. Extending Theorem 5.3 and Theorem 5.4 require some modification to the proofs. Nevertheless, more general theorem statements have recently been shown in [73, Theorems 1 and 2] and we refer the readers to [73] for details.

