# OpenReview forum: "HARD: Hyperplane ARrangement Descent"
_CPAL.cc/2024/Conference — CPAL 2024 (Proceedings Track) Oral_

### Official Review · Reviewer_m9My · 2023-10-02
**Report on HARD: Hyperplane ARrangement Descent**

**Rating:** 7
**Confidence:** 5

**Review:**

In this paper, the authors address the hyperplane clustering problem in the presence of outliers. They introduce the Hyperplane Arrangement Descent (HARD) algorithms and provide a comprehensive theoretical analysis. In my view, this work represents a significant accomplishment.

Comments:

1. It would be beneficial to include specific details regarding the computation of $b_k^{(t+1)}$ and elaborate on their computational complexity. Additionally, a comparative analysis of the computational complexity of your proposed method with other approaches based on different loss functions would enhance the paper's clarity.

2. Figure 2 needs further clarification. Please define the x-axis to ensure its meaning is clear to readers. Furthermore, could you explain why the running time of the red lines is higher at $K=4$ than at $K=5$ in Figure 2(c)?

---

### Official Review · Reviewer_vpD9 · 2023-10-04
**Solid work**

**Rating:** 8
**Confidence:** 3

**Review:**

This paper proposed a new algorithm for Hyperplane (2D subspace) clustering algorithm called hyperplane arrangement descent (HARD) that is both efficient and robust to outliers. This new family of algorithms solve a robust version of the Generalized Principal Component Analysis (GPCA) objective that uses a L1 norm for improved robustness against outliers, this new objective is called GPCA-l1.

This proposed GPCA-l1 objective can be optimized by a coordinate-wise algorithm called HARD-l1, however, this algorithm contains an inner optimization step that itself is cumbersome to optimize. Therefore the authors proposed to relax the inner optimization problem to a smoothed upper bound that can be efficiently solvable by SVD. The resulting algorithm is called HARD-l1+. The authors then study a smoother version of the GPCA-l1 that is called GPCA-Huber. Similar to HARD-l1, the authors relaxed the inner optimization of HARD-Huber to obtain a faster algorithm HARD-Huber+.

The authors move on to prove that the ground truth hyperplane arrangement minimizes the GPCA-l1 objective, and that the HARD-l1, HARD-Huber and HARD-Huber+ algorithm can successfully find critical points of GPCA-l1, GPCA-Huber and GPCA-Huber+ respectively at convergence. However, the authors find it difficult to prove that the practical HARD-Huber+ algorithm converges to a critical point.

Lastly, the authors used synthetic datasets to show that HARD-l1+ and HARD-Huber+ achieves better clustering accuracy as well as significantly superior runtime compared to other subspace or hyperplane clustering algorithms. The new algorithms also outperform previous algorithms on a more difficult and realistic clustering problem involving latent embedding learned by a self-supervised learning algorithm on CIFAR-10.

Overall, the paper is very well written and clear, the logic is easy to follow. Although the practical algorithms don't have theoretical guarantees, their benefit is clearly demonstrated. I look forward to seeing more applications of those algorithms in the ML community.

---

### Official Review · Reviewer_qtDd · 2023-10-08
**The theory is interesting and educational as it connects GPCA and DPCP formalisms. Though, the motivation is weak, the writing and the structure of the paper can be greatly improved, and detailed discussions are missing.**

**Rating:** 5
**Confidence:** 4

**Review:**

This paper studies the problem on subspace clustering. The main emphasis of the work are (1) hyperplane clustering, (2) using cost functions that enable robustness to outlier data, namely, L1 and Huber and their variations, (3) providing theoretical guarantees on various notions of the optimality of estimated hyperplanes ---  under strong conditions on data points and outliers.  The experiments in the main text include numerical results on small-scale synthetic data and CIFAR-10 features.

####################################################################
My evaluation:
This paper introduces interesting theoretical results that combine GPCA formulation of hyperspheres with robust L1 clustering cost (inspired by DPCP) and prove certain optimalities for estimated hyperspaces under strong geometric conditions. However, this does not qualify to be accepted due to the following issues: (a) The motivation of using "only" hyperspaces is not fully explained. What happens when we have subspaces of codimensions >2? (b) The writing the paper is poor : unscientific word and claims, the structure, explanation of the assumptions 5.1 is insufficient, ...  (c) The experiments includes small-scale data. What happens when we have no outliers or we have different variances for random additive noises? You can should move some of the "no-outlier" results in the appendix to the main text and investigate how this method compares in estimating the correct subspace with other methods (in the no-outlier scenario).

P.S. I assume that the proposed Theorems are correct as I did not read proofs in the Appendix.
####################################################################
I will summarize my main questions, comments, and suggestions as follows:

(1) Regarding the "Geometric Quantities": Suppose datapoints belong to subspace of codimension 2 (a subspace of the supposed hyperplane). Then, it is easy to show that c_{in, k , min} is 0 for all k \in [K]. In this case, we are not guaranteed to have {b^{*}_k} as coordinate-wise minimizers of (GPCA-l1) --- according to Theorem 4.3. The authors should explain what happens in this situation --- which is quite common in practice.

(2) Related to my comment (1), the proposed method formalizes the problem of subspace clustering using orthogonal complement representation of the hyperspaces, that is, vectors { b_k }. Therefore, once these vectors estimated, the algorithm returns the estimated hyperspaces. One can easily think of a situation where we want to estimate high-dimensional subspaces with codimensions > 1. What would be the appropriate modification of this algorithm to operate on those cases? One may think that you have to resort to estimating one-dimension at a time in the orthogonal complement space --- which would be greedy. It is important to clearly explain your approach from this viewpoint so that the reader can easily compare this algorithm with other methods.

(3) Assumption 5.1 might be standard but not realistic at all. Either the authors should provide their insight on why (or on which class of point sets) this assumption is true or clearly explain that this assumption puts a strong set of constraints on both inlier and outlier point sets -- which maybe even impossible to satisfy. Regarding (A3), please either prove the claim that for random points the eigenvectors of the weighted sum are distinct (with probability 1) or cite a reference.

(4) (page 2, line 43) 	“KH-DPCP [27] integrates DPCP into the K-Hyperplanes framework. “ + (page 3, line 102) “Second, the underlying theory of why such an integration works well has thus far remained, to our knowledge, obscure. “  The algorithm KH-DPCP is not discussed in the main text. Given the fact that its objective is the most comparable to this problem  (robust hyperplane clustering), it is beneficial to dedicate a detailed remark on this algorithm.

In what follows, I give comments on issues that appear frequently in the paper (not limited to these examples).

(5) The specific choice of notations may seem irrelevant. But from the perspective of readers, authors should try to use “intuitive” notations. Two examples (among many): (a) In equation (GPCA-l2), indices in summation goes from j=1 to N. why not n=1 to N? (b) The definition of w^{t}_{j,k} in Algorithm 1: HARD-l is "similar" to that of d^{k}_j in subsection 4.1. (c) A good notation can simplify Equation (2) and make it more intuitive, namely, I - bb^{T} is an orthogonal projection matrix ...

(6) (page 1, line 24) “However, for subspaces of high relative dimensions (relative to D), sparsity and low-rankness break down, and so do these methods.” The word relative is repetitive. The sentence is broken.

(7) (page 1, line 36)  “It is very simple and intuitive, but it is inaccurate and not robust to outliers, and it has limited theoretical guarantees (e.g., of convergence to true hyperplanes). “  Please do not use words that do not convey useful information, like “very”. Please also do not construct run on sentences ( … but … and  … and  … ). This hinders the readability of your paper.

(8) (page 2, line 48) 	“we blend the GPCA and DPCP philosophies”  I’m not sure if “philosophies” is an appropriate word here.

(9) (page 2, line 44) “In doing so, it inherits the one-shot ability of K-Hyperplanes and the robustness of DPCP to a certain extent. But it also compromises accuracy and comes with no theoretical guarantees”
“to a certain extent” does not make sense to me. Could you please explain this algorithm, and compare it with your proposed algorithms?

(10) (page 3, line 92) “The first idea of [27] has some  … ” Please do not use a numerical reference as an object in a sentence. You can use the authors’ names instead.

(11) (page 4, line 127) “How can we solve the harder problem (GPCA-l1) efficiently?“ I am not sure in what sense the problem is hard. Maybe instead you can emphasize on the fact that the l1 objective is nonsmooth?

(12) (page 4, line 132) Whether the true normal vectors are a global minimizer of (GPCA-l1)? I do not understand the meaning of this sentence.

(13) (page 7, line 229) "Intuitively, (A2) is more likely than (A1) to have a unique global minimizer because h is further locally quadratic (strongly convex)." The term likely implies the existence of an underlying probability distribution. If there in none, this is an incorrect statement.

---

### Meta-Review · Area_Chair_kq2G · 2023-11-07

**Recommendation:** Accept (Oral)
**Confidence:** 5

**Metareview:**

In this work, the authors propose Hyperplane ARrangement Descent (HARD), a novel method for clustering points on high-dimensional subspaces. HARD learns multiple hyperplanes simultaneously through a non-convex, non-smooth L1 minimization problem, outperforming existing methods and demonstrating its efficacy in clustering deep features on CIFAR-10.

This paper has received generally positive feedback from two reviewers, indicating its potential and merit. However, one reviewer expressed concerns regarding the motivation presented in the paper, as well as specific writing details. Upon thorough examination of the reviewers' comments and the authors' responses in the rebuttal, it is evident that the authors have made commendable efforts to address the concerns raised. They have provided coherent explanations and clarifications, demonstrating a willingness to enhance the paper based on the reviewers' feedback.

Considering the authors' diligent response and the potential significance of the paper, I recommend $\textbf{acceptance}$, provided that the authors make the necessary revisions as outlined in their rebuttal. These revisions should focus on strengthening the paper's motivation and addressing the specific writing concerns raised by the reviewers.

Finally, the existing convergence analysis result is relatively limited, ensuring only convergence to critical points or minimizers. It is imperative for the authors to analyze convergence to the true hyperplanes.

---

### Decision · Program_Chairs · 2023-11-19

**Decision:**

Accept (Oral)

**Comment:**

All reviewers and AC agreed that the paper is of high quality. In this work, the authors propose Hyperplane ARrangement Descent (HARD), a novel method for clustering points on high-dimensional subspaces. HARD learns multiple hyperplanes simultaneously through a non-convex, non-smooth L1 minimization problem, outperforming existing methods and demonstrating its efficacy in clustering deep features on CIFAR-10. On the other hand, the existing convergence analysis result is still limited, ensuring only convergence to critical points or minimizers. It would be more impressive if the authors could show convergence to the true hyperplanes.

The action PC chair for this paper is Qing Qu, who made the decision after carefully reading the paper as well as the comments by all reviewers and AC. The decision is agreed upon by all PC chairs.